# HUWE1 E3 ligase promotes PINK1/PARKIN-independent mitophagy by regulating AMBRA1 activation via IKKα

Anthea Di Rita[1,2,14], Angelo Peschiaroli[3], Pasquale D'Acunzo [2], Daniela Strobbe[1,4], Zehan Hu[5], Jens Gruber[6], Mads Nygaard [7], Matteo Lambrughi[7], Gerry Melino[8], Elena Papaleo[7], Jörn Dengjel [5], Said El Alaoui[9], Michelangelo Campanella[4,10,11], Volker Dötsch[6], Vladimir V. Rogov [6], Flavie Strappazzon[1,12] & Francesco Cecconi[1,2,13]

The selective removal of undesired or damaged mitochondria by autophagy, known as mitophagy, is crucial for cellular homoeostasis, and prevents tumour diffusion, neurodegeneration and ageing. The pro-autophagic molecule AMBRA1 (autophagy/beclin-1 regulator-1) has been defined as a novel regulator of mitophagy in both PINK1/PARKIN-dependent and -independent systems. Here, we identified the E3 ubiquitin ligase HUWE1 as a key inducing factor in AMBRA1-mediated mitophagy, a process that takes place independently of the main mitophagy receptors. Furthermore, we show that mitophagy function of AMBRA1 is post-translationally controlled, upon HUWE1 activity, by a positive phosphorylation on its serine 1014. This modification is mediated by the IKKα kinase and induces structural changes in AMBRA1, thus promoting its interaction with LC3/GABARAP (mATG8) proteins and its mitophagic activity. Altogether, these results demonstrate that AMBRA1 regulates mitophagy through a novel pathway, in which HUWE1 and IKKα are key factors, shedding new lights on the regulation of mitochondrial quality control and homoeostasis in mammalian cells.

---

[1] Department of Biology, University of Rome Tor Vergata, 00133 Rome, Italy. [2] Department of Paediatric Haematology, Oncology and Cell and Gene Therapy, IRCCS Bambino Gesù Children's Hospital, Rome, Italy. [3] National Research Council of Italy (CNR), Institute of Translational Pharmacology IFT, Via Fosso del Cavaliere 100, 00133 Rome, Italy. [4] IRCCS- Regina Elena, National Cancer Institute, 00133 Rome, Italy. [5] Department of Biology, University of Fribourg, Fribourg, Switzerland. [6] Institute of Biophysical and Center for Biomolecular Magnetic Resonance, Goethe University, Frankfurt, Germany. [7] Computational Biology Laboratory, Danish Cancer Society Research Center, 2100 Copenhagen, Denmark. [8] Department of Experimental Medicine and Surgery, University of Rome Tor Vergata, 00133 Rome, Italy. [9] Covalab, Villeurbanne, France. [10] Department of Comparative Biomedical Sciences, Royal Veterinary College, London NW1 0TU, UK. [11] University College London Consortium for Mitochondrial Research, University College London, London WC1 6BT, UK. [12] IRCCS FONDAZIONE SANTA LUCIA, 00143 Rome, Italy. [13] Unit of Cell Stress and Survival, Danish Cancer Society Research Center, 2100 Copenhagen, Denmark. [14] Present address: IRCCS FONDAZIONE SANTA LUCIA, 00143 Rome, Italy. Correspondence and requests for materials should be addressed to F.S. (email: f.strappazzon@hsantalucia.it) or to F.C. (email: cecconi@cancer.dk)

Mitophagy is an evolutionary-conserved mechanism that allows damaged or undesired mitochondria removal by an autophagosome–lysosome pathway[1]. This high-quality clearance system is fundamental for preserving cellular homoeostasis and for critical processes, such as inflammation and cell death or diseases, including cancer and neurodegeneration[2]. The main mitophagy pathway is driven by the stabilization of the mitochondrial kinase PINK1, resulting in the recruitment of the E3 ubiquitin ligase PARKIN to damaged mitochondria, and in a ubiquitylation cascade targeting several outer mitochondrial membrane (OMM) proteins. Indeed, ubiquitylation events are fundamental during mitophagy, contributing to the normal turnover of mitochondrial proteins in basal conditions[3], and promoting the recognition of UBD (ubiquitin binding domain)-containing proteins, which allow mitochondria selective autophagy[4]. Optineurin (OPTN) and NDP52 are the the main mitophagy receptors, acting as bridges between ubiquitin-tagged mitochondria and the autophagosome-associated protein, MAP1LC3/LC3 (microtubule-associated proteins 1A/1B light chain 3), thus leading to mitochondria engulfment into autophagosomes, upon mitochondrial membrane depolarization[5,6]. More recently, PHB2 (Prohibitin-2), an inner mitochondrial membrane protein, has also been demonstrated to be involved in selective mitochondria removal, cooperating with PARKIN in mammals[7].

Besides the PINK1/PARKIN system, the OMM proteins NIX/BNIP3L, Bcl2-L-13 and FUNDC1 are also fundamental to trigger mitophagy in mammals, by interacting directly with LC3 and regulating mitochondrial clearance. In particular, (i) NIX/BNIP3L promotes mitochondria removal during reticulocytes differentiation[8,9]; (ii) Bcl2-L-13 is the mammalian homologue of Atg32, and it stimulates mitochondria fragmentation and therefore mitophagy[10]; and lastly, (iii) FUNDC1 allows mitochondrial clearance upon hypoxia[11]. Of note, these mitophagy receptors are post-translationally modified in order to regulate their interaction with LC3 during mitophagy[12].

We previously demonstrated that the LC3-interacting protein AMBRA1 plays a role in the selective degradation of ubiquitylated mitochondria, transducing both canonical PINK1/PARKIN-dependent and -independent mitophagy[13]. Here, we describe HUWE1 as the novel E3 ubiquitin ligase that collaborates with AMBRA1 to induce mitochondrial clearance, by inducing mitofusin 2 (MFN2) degradation. Moreover, since AMBRA1 exhibits (i) mitochondria localization[14], (ii) a LIR (LC3-interacting region) motif and (iii) the capacity to induce mitophagy[13], we decided to investigate whether AMBRA1 could be defined as a receptor, and to better characterize this pathway. We thus found that the activity of the mitophagy receptor AMBRA1 is regulated by a phosphorylation upstream of its LIR motif, mediated by the IKKα kinase.

Altogether, these findings highlight AMBRA1, HUWE1 and IKKα as three novel and crucial proteins for mammalian mitophagy regulation, following mitochondrial membrane depolarization in a PINK1/PARKIN-free context.

## Results

**HUWE1 is required for AMBRA1-mediated mitophagy.** Since we have previously shown that AMBRA1 regulates the dismissal of ubiquitylated mitochondria in PINK1/PARKIN-independent mitophagy[13], we searched for a novel putative E3 ubiquitin ligase, which could control mitochondrial protein ubiquitylation in cooperation with AMBRA1 during the mitophagy process.

To this aim, we performed a SILAC (stable isotope labelling by amino acids in cell culture)-based mass spectrometry (MS) analysis in order to detect AMBRA1-interacting proteins, upon mitophagy stimulation, in a PARKIN-free cellular system. Thus, we immunoprecipitated Myc-AMBRA1$^{ActA}$ (an AMBRA1 fusion protein targeted to the external membrane of mitochondria), which stimulates mitophagy[13] in HeLa cells grown in two different isotope labelling media (light and heavy). The immunoprecipitated samples of the negative control (light medium lysate) and the experimental sample (heavy medium lysate) were mixed and then analysed by MS analysis[15]. Interestingly, this screening led us to identify a single E3 HECT-Ubiquitin ligase, HUWE1 (ARF-BP1, MULE, UREB1), whose role in macroautophagy/mitophagy was yet undisclosed (Fig. 1a).

In order to confirm these MS results, we performed a co-immunoprecipitation experiment in HeLa cells expressing a vector encoding for Myc-AMBRA1$^{ActA}$. Total lysates were immunoprecipitated with an anti-Myc antibody or an IgG control and the immune-complexes were analysed by western blotting. We found that AMBRA1$^{ActA}$ was able to co-immunoprecipitate with HUWE1 (Fig. 1b). Since the expression of AMBRA1$^{ActA}$ induces mitophagy per se[13], we decided to compare the binding between AMBRA1$^{WT}$ and HUWE1 in basal versus mitophagy conditions. To this end, we over-expressed HUWE1- and Myc-AMBRA1$^{WT}$-encoding constructs in HeLa cells, treated or not with oligomycin and antimycin (O/A) in order to induce mitophagy. Total lysates were immunoprecipitated using an anti-HUWE1 antibody or IgG control and the immune-complexes were analysed by western blot. As shown in Fig. 1c, AMBRA1$^{WT}$ preferentially binds HUWE1 following mitophagy induction.

Given the evidence that HUWE1 is an AMBRA1-interacting protein preferentially upon mitophagy stimulation, we then focussed on a putative role of HUWE1 in mitophagy.

First, since HUWE1 is an E3 ubiquitin ligase mainly found in the cytoplasm[16], we checked whether it was present at the mitochondria following mitophagy induction. By performing a mitochondria–cytosol fractionation in HeLa cells expressing Myc-AMBRA1$^{ActA}$ or an empty vector (PcDNA3), we found that AMBRA1$^{ActA}$ expression enables HUWE1 localization at the mitochondria (Fig. 1d). It is noteworthy that HUWE1 translocates to the mitochondria in also AMBRA1$^{WT}$-expressing cells upon mitophagy stimulation (Fig. 1e).

Next, in order to assess the effect of HUWE1 genetic silencing on AMBRA1$^{ActA}$-induced mitophagy, we checked mitophagy progression upon downregulation of HUWE1 via small interfering RNA (siRNA). As shown in Fig. 1f and in Supplementary Figure 1a, the decrease of mitochondrial markers associated with AMBRA1$^{ActA}$ expression[13], such as COXII and COXIV, is rescued in HUWE1-interfered cells. To confirm these data, we performed a confocal microscopy analysis in cells co-expressing both GFP-ShHUWE1 or GFP-ShCtr together with AMBRA1$^{ActA}$, and we measured the mitochondria amount in transfected green fluorescent protein (GFP)-positive cells. As expected, following AMBRA1$^{ActA}$ expression with GFP-ShCtr, we observed the formation of mitochondria aggregates (mito-aggresomes) around the nuclei, and a decrease in mitochondria staining; by contrast, GFP-ShHUWE1 transfected cells presented a delay in both mito-aggresomes formation and mitochondrial clearance (Supplementary Fig. 1b). In addition, since we had previously demonstrated that AMBRA1 is able to induce mitophagy in a PINK1-independent manner[13], we analysed AMBRA1–HUWE1 effect in mitophagy regulation also in mt-mKeima-expressing PINK1 knockout (mt-mKeima-PINK1-KO) HeLa cells[5]. As shown in Supplementary Fig. 1c, besides a strong localization of mitochondria into an acidic environment, following Myc-AMBRA1$^{ActA}$ transfection, we also observed a reduction of acidic mt-mKeima upon HUWE1 downregulation. These data confirm that HUWE1

depletion impairs AMBRA1-mediated mitophagy. Finally, we also confirmed these data with the wild-type AMBRA1, following mitophagy induction. In fact, we transfected HeLa cells with Myc-AMBRA1$^{WT}$ in combination or not with siRNA-HUWE1. Then, we treated cells with O/A (2.5 μM and 0.8 μM, respectively, 4 h). By western blot analysis, we observed a decrease of mitochondrial markers, COXII and COXIV, in the mitophagy context (Fig. 1g and Supplementary Fig. 1c).

In sum, these findings define a central role for the E3 ubiquitin ligase HUWE1 in controlling AMBRA1-dependent mitophagy.

**Upon mitophagy, AMBRA1 acts as a cofactor for HUWE1 activity.** Since HUWE1 depletion results in a block of AMBRA1-mediated mitophagy, we hypothesized that, in the meantime, mitochondria could be less ubiquitylated in that context.

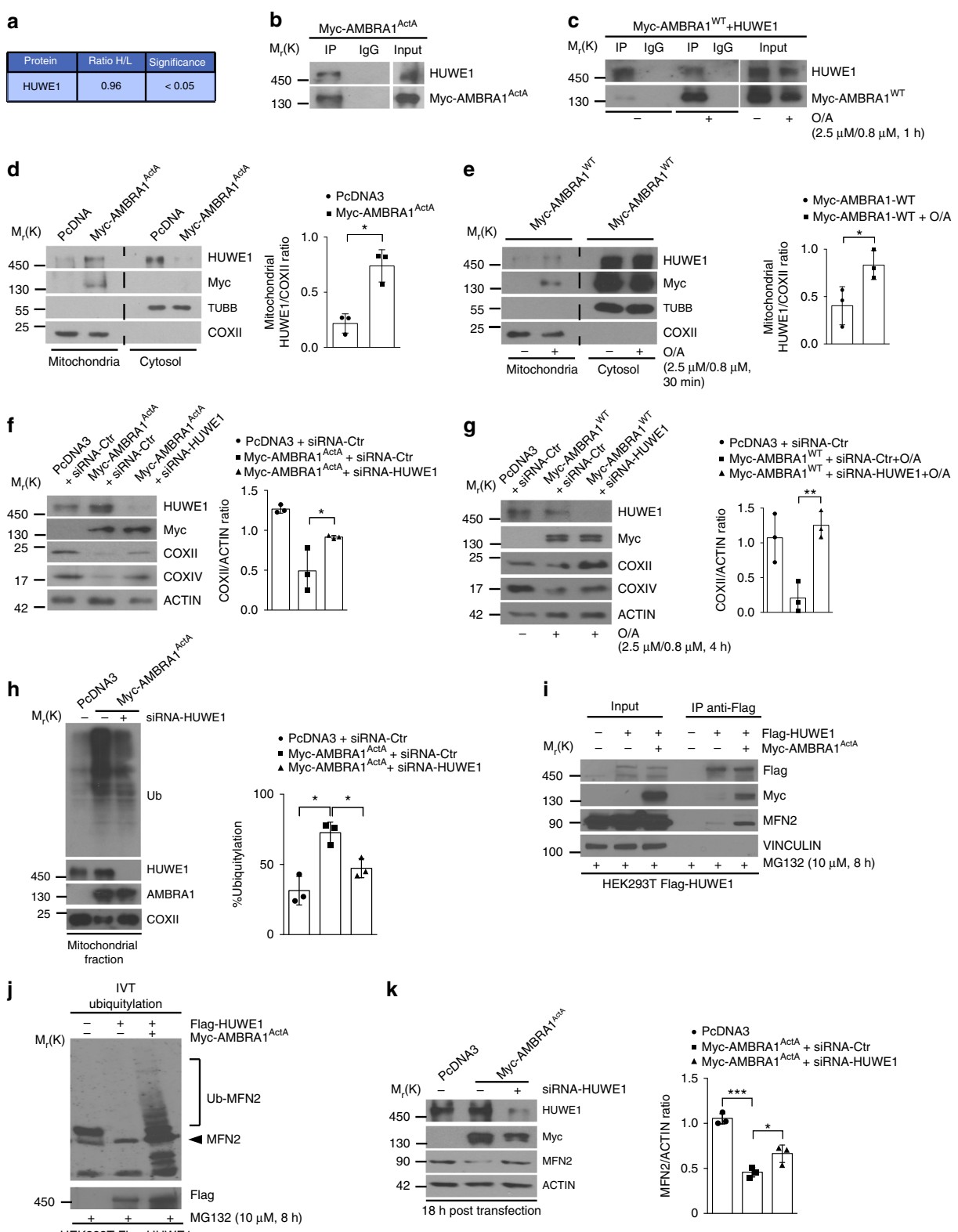

To verify this hypothesis, we transfected HeLa cells with Ctr-siRNA or HUWE1-siRNA in combination with a vector encoding Myc-AMBRA1[ActA]. We then performed a subcellular fractionation and analysed ubiquitylation in the mitochondria lysates, measuring the amount of total ubiquitin in these extracts, following AMBRA1[ActA] transfection. As expected, we observed a reduction of the total mitochondrial ubiquitylation in HUWE1-interfered cells during AMBRA1[ActA]-induced mitophagy when compared to control cells (Fig. 1h).

We next focalized our attention on MFN2, an OMM pro-fusion protein, whose ubiquitylation is known to be mediated by several E3 ubiquitin ligases, including HUWE1[17], and whose degradation is known to be crucial for mitophagy induction[18]. We thus tested whether HUWE1–MFN2 interaction could be regulated by AMBRA1, upon mitophagy induction. To this aim, we transfected HEK293T cells stably expressing Flag-HUWE1 with a vector encoding Myc-AMBRA1[ActA]. After treatment with the proteasome inhibitor MG132, we immunoprecipitated cell lysates with an anti-Flag antibody and observed that the expression of Myc-AMBRA1[ActA] strongly favours the interaction between HUWE1 and MFN2 (Fig. 1i). Accordingly, by an in vitro ubiquitylation assay, we observed that Myc-AMBRA1[ActA] expression in HEK293T-Flag-HUWE1 cell clones strongly increases MFN2 ubiquitylation compared to control conditions (Fig. 1j).

Last, we checked for MFN2 degradation in AMBRA1-mediated mitophagy (18 h following Myc-AMBRA1[ActA] expression). By analysing MFN2 protein levels in Myc-AMBRA1[ActA] transfected cells, we observed a decrease in this OMM protein. Interestingly, MFN2 decrease is rescued by downregulating HUWE1 or by treating cells with MG132 (Fig. 1k and Supplementary Fig. 1d). These data suggest that MFN2 ubiquitylation is an early event, following Myc-AMBRA1[ActA] expression, controlled by HUWE1 and most likely required for AMBRA1-mediated mitophagy.

Altogether, these results indicate that AMBRA1 favours MFN2–HUWE1 interaction, thus leading to MFN2 ubiquitylation and subsequent degradation, with a final effect on mitophagy induction. AMBRA1 can be thus considered as a novel cofactor for HUWE1 ubiquitin ligase activity, underlining a new paradigm for the regulation of HUWE1.

**AMBRA1 is a relevant mitophagy receptor**. Next, since AMBRA1 (i) operates as a cofactor for HUWE1 activity in order to trigger mitophagy, (ii) displays mitochondrial localization[14], (iii) exhibits a LIR motif on its sequence for LC3B binding, (iv) is able to regulate PINK1/PARKIN-independent mitophagy and (v) controls the PINK1/PARKIN-mediated mitophagy[13], we investigated whether AMBRA1 can act as a mitophagy receptor.

To this aim, we checked for AMBRA1-mediated mitophagy efficiency using the well-known model of stable Penta-KO cells, unable to perform mitophagy[5]. Indeed, these cells are deficient for five mitophagy receptors, i.e., OPTN, NDP52, NBR1, P62 and TAX1BP1. In these cells, only the expression of NDP52 or OPTN (or TAX1BP1, in a few cases) is able to rescue the mitophagy pathway. In fact, the treatment of these cells with mitochondrial uncouplers does not induce mitophagy (Supplementary Fig. 2a). Thus, we transfected Penta-KO cells with a vector encoding Myc-AMBRA1[ActA]. Western blot analysis, performed on extracts obtained 24 h after transfection, showed a decrease of mitochondrial markers, i.e., COXII, COXIV, and also HSP60 (Supplementary Fig. 2b), underlining that AMBRA1 expression at the mitochondria is sufficient per se to induce mitophagy in Penta-KO cells (Fig. 2a). Furthermore, by performing a confocal microscopy analysis, we observed that AMBRA1[ActA] expression in Penta-KO cells led to mitochondria perinuclear re-localization, typical of mitophagy induction, followed by a decrease in the mitochondrial mass (Fig. 2b).

Given the powerful ability of AMBRA1[ActA] to induce mitophagy in these cells, we also checked whether wild-type AMBRA1 could promote mitochondria selective removal. To this end, we transiently transfected Penta-KO cells with a vector encoding for Myc-AMBRA1[WT] and treated cells with O/A for 24 h in order to induce mitophagy. By western blot analysis, we observed a decrease in the levels of the mitochondrial markers COXII and COXIV, thus confirming the central role of AMBRA1 in promoting mitophagy in this cell line (Fig. 2c). Of note, following overexpression of AMBRA1[WT], in the absence of any mitophagy treatments, all mitochondrial markers showed a marked decrease.

Also in this case, we confirmed the data by performing a confocal microscopy analysis in which we observed the formation of mito-aggresomes and then a decrease of mitochondria content in cells overexpressing AMBRA1[WT] (Fig. 2d). Finally, in order to ascertain that the AMBRA1-dependent mitochondrial decrease was well associated to lysosomal degradation, we treated cells with the lysosomal inhibitor $NH_4Cl$ (Supplementary Fig. 2c). Indeed, an almost complete rescue of COXIV and COXII levels by $NH_4Cl$ confirmed that a block in lysosomal degradation restored the total amount of mitochondria within the cells.

Taken together, our findings reveal that AMBRA1 acts as a relevant mitophagy receptor, capable to induce mitophagy in NDP52-, OPTN-, TAX1BP1-, NBR1- and P62-deficient cells.

**AMBRA1 phosphorylation regulates AMBRA1-dependent mitophagy**. A key process in the induction of selective autophagy is the interaction between mitophagy receptors and Atg8 (autophagy-related protein 8) protein family members[19], which

**Fig. 1** HUWE1 cooperates in AMBRA1-mediated mitophagy. **a** Table showing the E3 ubiquitin ligase interacting with AMBRA1 upon mitophagy stimuli in HeLa PARKIN-free cells. The ratio of heavy labelled peptides to the remaining non-labelled ones (ratio H/L) reflects is indicated. **b** Representative image of AMBRA1[ActA]-HUWE1 co-immunoprecipitation (Co-IP); n = 3. **c** Representative image of AMBRA1[WT]-HUWE1 co-immunoprecipitation (Co-IP) upon mitophagy stimulation (O/A); n = 3. **d** PcDNA3 or Myc-AMBRA1[ActA] transfected HeLa cells were subjected to mitochondria–cytosol purification. Samples were blotted for the indicated antibodies; n = 3. **e** PcDNA3 or Myc-AMBRA1[WT] transfected HeLa cells were treated with O/A for 30 min and subjected to mitochondria–cytosol purification; n = 3. **f** HeLa cells transfected with PcDNA3 or Myc-AMBRA1[ActA] vectors were transfected with siRNA-Ctr or siRNA-HUWE1 constructs and then immunoblotted for the indicated proteins; n = 3. **g** Following O/A treatment, PcDNA3- or Myc-AMBRA1[WT]-expressing HeLa cells transfected with siRNA-Ctr or siRNA-HUWE1 were immunoblotted for the indicated antibodies; n = 3. **h** PcDNA3, Myc-AMBRA1[ActA] and Myc-AMBRA1[ActA] transfected cells interfered for HUWE1 were blotted for the indicated antibodies. The graph shows the total ubiquitin amount in mitochondria fractionation sample (%); n = 3. **i** HEK293T stably expressing Flag-HUWE1, transfected with a vector coding for Myc-AMBRA1[ActA] and treated with MG132 (10 μM, 8 h) were co-immunoprecipitated and immunoblotted for the indicated proteins; n = 3. **j** In vitro ubiquitylation assay of cells in **i**. Protein samples were resolved by SDS–PAGE and immunoblotted for Myc, MFN2 and HUWE1; n = 2. **k** HeLa cells transfected with Myc-AMBRA1[ActA] (18 h) and interfered for HUWE1 (siRNA-HUWE1) were analysed by western blot; n = 3. The quantification results are the mean of three independent experiments (±S.D.). *P < 0.05; **P < 0.01. Statistical analysis was performed using one-way ANOVA (**f**, **g**, **k**) and Student's t-test (**d**, **e**, **h**). $M_r(K)$ = relative molecular mass expressed in kilodalton

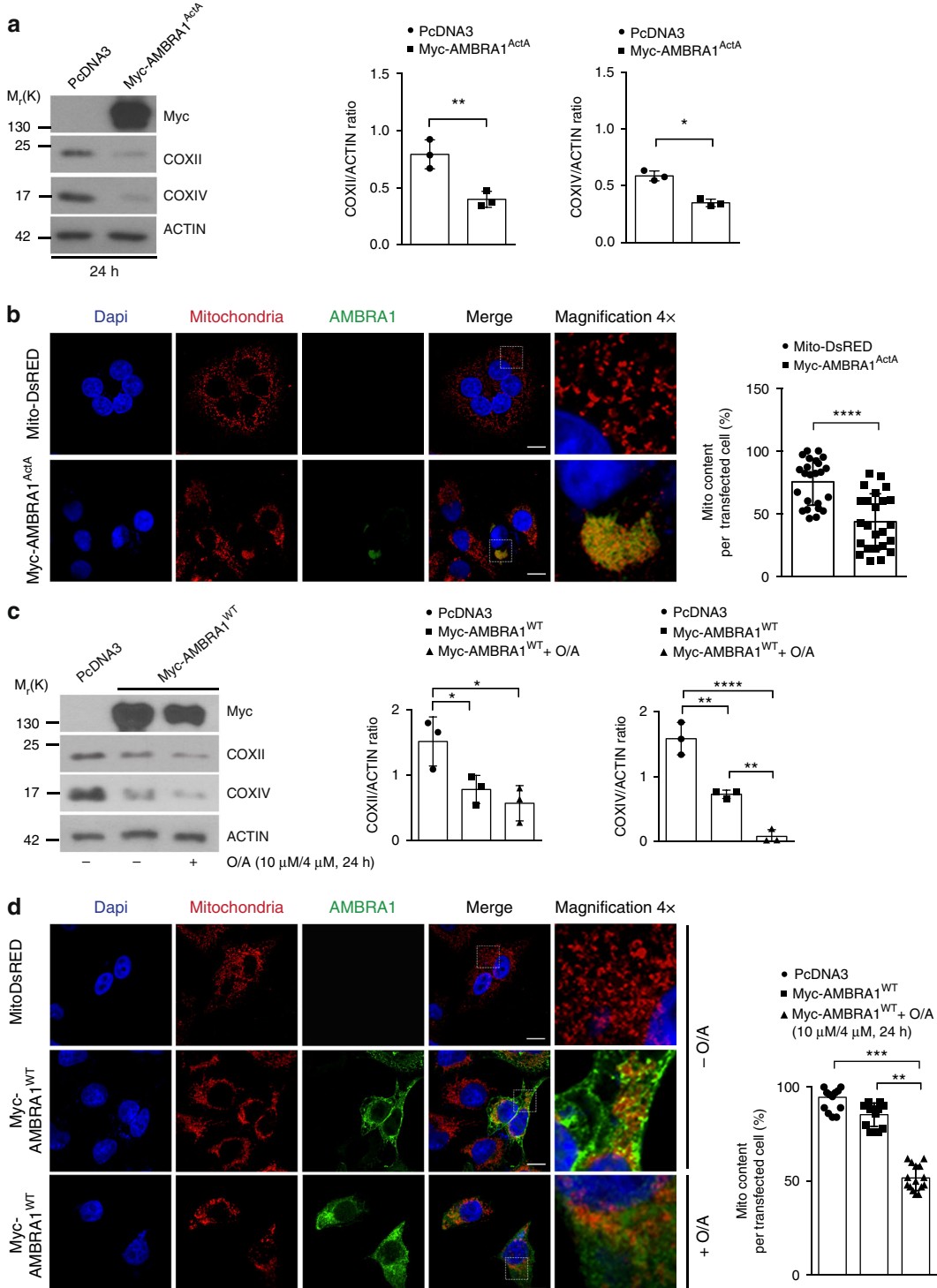

**Fig. 2** AMBRA1 expression rescues mitophagy induction in Penta-KO cells. **a** Penta-KO cells transfected with a vector encoding for Myc-AMBRA1[ActA] or an empty vector (PcDNA3) for 24 h were immunoblotted for the indicated antibodies; $n = 3$. **b** Myc-AMBRA1[ActA] (red) or Mito-DsRED (red) transfected Penta-KO cells were fixed after 24 h. Scale bar, 10 μm. The graph shows the mito content (%) in single transfected cells (±S.D.); $n = 4$. Mito-DsRED = 12 individual fields; Myc-AMBRA1[ActA] = 12 individual fields. **c** Penta-KO cells were transfected with vectors encoding Myc-AMBRA1[WT] or PcDNA3 and treated with O/A (10 μM and 4 μM, respectively) for 24 h. Mitochondrial markers levels, COXII and COXIV, were evaluated by western blot; $n = 3$. **d** Cells transfected with Mito-DsRED or with Myc-AMBRA1[WT] were immunostained with an anti-Myc (green) and anti-TOM20 (red) antibodies. Scale bar, 10 μm; $n = 3$. Mito-DsRED = 9 individual fields; Myc-AMBRA1[WT] = 9 individual fields; Myc-AMBRA1[WT]+O/A = 9 individual fields. *$P < 0.05$; **$P < 0.01$; ***$P < 0.001$; ****$P < 0.0001$. The quantification results are the mean of three independent experiments (±S.D.) if not otherwise stated. Statistical analysis was performed using Student's $t$-test (**a**, **b**) or one-way ANOVA (**c**, **d**). $M_r(K)$ = relative molecular mass expressed in kilodalton

are LC3 and GABARAP proteins in mammals (mATG8). Since the phosphorylation status of a serine residue (Ser) flanking the LIR motifs of known mitophagy receptors, such as FUNDC1[11], BNIP3[20], NIX/BNIP3L[9,21], Bcl2-L-13[10] and OPTN[22], controls their binding with LC3, we investigated whether AMBRA1 could be post-translationally modified in a serine site in proximity of its LIR motif[13].

First, through an in silico analysis of the AMBRA1 sequence, we identified serine 1014 (S1014) as a putative phosphorylation site close to the AMBRA1-LIR motif (Fig. 3a). Second, in order to demonstrate whether the phosphorylation on S1014 was able to influence AMBRA1–LC3B interaction in cells, we generated two AMBRA1 mutants, a phospho-dead mutant (AMBRA1^{S1014A}) and a phospho-mimetic mutant (AMBRA1^{S1014D}). By

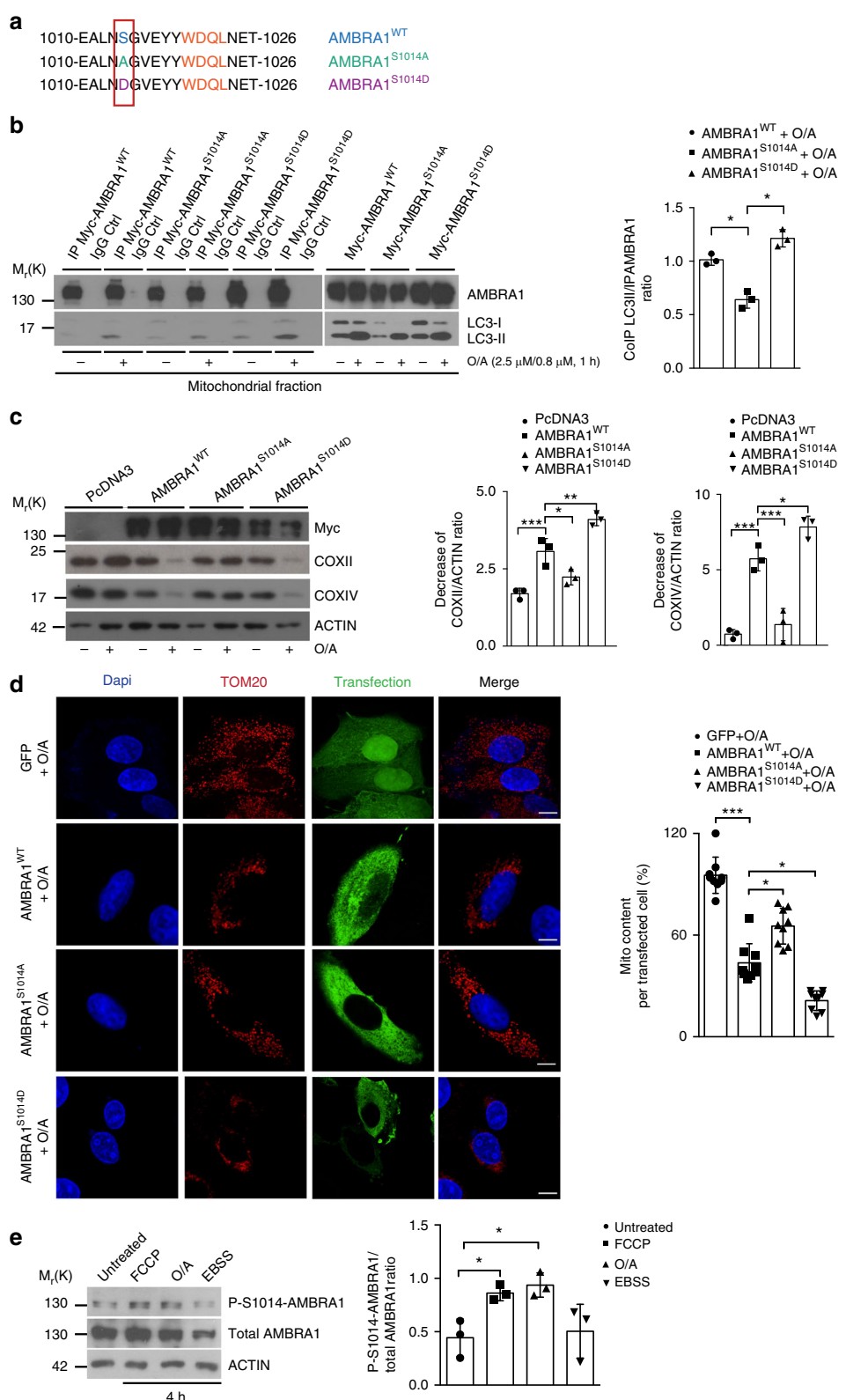

performing co-immunoprecipitation experiments, we analysed AMBRA1–LC3B interaction in basal conditions and upon mitophagy stimulation (O/A treatment), obtained by overexpressing Myc-AMBRA1[WT] and AMBRA1[S1014A] or AMBRA1[S1014D] in HeLa cells. First, we confirmed that AMBRA1 interacts mostly with LC3B following mitophagy stimuli[13]. Moreover, we observed that alanine substitution at residue 1014 reduced the binding between AMBRA1 and LC3B, following O/A treatment (Fig. 3b). By contrast, the AMBRA1[S1014D] enhanced the ability of AMBRA1 to interact with LC3B during mitophagy.

These results highlight that the phosphorylation status of AMBRA1 on S1014 influences the interaction between AMBRA1 and LC3B during mitophagy.

Next, in order to ascertain whether S1014 is relevant in AMBRA1-induced mitophagy, we performed functional experiments to evaluate the mitophagy activity of AMBRA1[WT] compared with its mutants on S1014. To this aim, we transfected HeLa cells with the empty vector PcDNA3 as a control, and vectors encoding Myc-AMBRA1[WT] or Myc-AMBRA1[S1014A] or Myc-AMBRA1[S1014D] and checked, by western blot analysis, the levels of mitochondrial proteins COXII and COXIV.

Interestingly, upon mitophagy stimulation (by O/A treatment), HeLa cells transfected with AMBRA1[WT] showed a decrease in COXII and COXIV levels, while cells expressing the AMBRA1[S1014A] mutant showed a lower decrease in mitochondrial marker levels compared with AMBRA1[WT] or AMBRA1[S1014D] (Fig. 3c). Of note, the AMBRA1[S1014D] mutant significantly increased the mitophagy ability of AMBRA1 when compared with the AMBRA1[WT] form (Fig. 3c). Also, we analysed mitochondrial clearance by confocal immunofluorescence, looking at TOM20 levels, in HeLa cells expressing GFP, Myc-AMBRA1[WT], Myc-AMBRA1[S1014A] or Myc-AMBRA1[S1014D]. Indeed, we observed that the phospho-mimicking mutation (Myc-AMBRA1[S1014D]) induced mito-aggresome formation and a clear decrease of mitochondria amount to a larger extent than AMBRA1[WT] (Fig. 3d). Moreover, we also validated these data using HeLa cells expressing the mito-mKeima vector[5]. In analogy with our biochemistry and immunofluorescence analysis, mitochondria of Myc-AMBRA1[S1014D]-expressing cells were predominantly found in an acidic environment, suggesting ongoing mitophagy. By contrast, Myc-AMBRA1[S1014A] transfected cells, in which we previously observed a delay in mitophagy, were mostly present in a neutral environment (Supplementary Fig. 3a).

Since we demonstrated that phosphorylation of AMBRA1 on S1014 was a crucial event in order to regulate AMBRA1-mediated mitophagy, we next decided to check whether endogenous AMBRA1 was also subjected to such modification. We thus generated, through rabbit immunization, an anti-AMBRA1 serum that specifically recognized the phosphorylation at S1014 (anti-Phospho(P)-S1014-AMBRA1 serum). We tested its efficiency by comparing the level of phospho-S1014 in basal versus mitophagy (carbonyl cyanide-4-(trifluoromethoxy)

phenylhydrazone (FCCP), O/A) or autophagy conditions (Earle's balanced salt solution (EBSS)). As shown in the graph (Fig. 3e), we observed a significant increase in the phosphorylation signal following FCCP and O/A treatments, indicating that phosphorylation of AMBRA1 on S1014 was occurring specifically upon mitochondrial membrane depolarization during mitophagy induction. In addition, we proved that the anti-P-S1014-AMBRA1 antibody was able to specifically recognize AMBRA1 by analysing endogenous AMBRA1 levels in siRNA-Ctr- or AMBRA1-interfered cells by siRNA-AMBRA1 (Supplementary Fig. 3b). We also confirmed that this antibody was able to better recognize over-expressed AMBRA1[WT] if compared to the phospho-dead AMBRA1[S1014A] form (Supplementary Fig. 3c).

Taken together, these data indicate that the phosphorylation of S1014 on AMBRA1 regulates the AMBRA1–LC3B interaction and its mitophagy activity. We have also generated an anti-P-S1014-AMBRA1 antibody that recognized this AMBRA1 dynamic post-translational modification.

**S1014-AMBRA1 phosphorylation increases the binding with mATG8s.** In order to characterize the binding affinity between AMBRA1-LIR and LC3/GABARAP proteins, and to confirm that the phosphorylation of S1014 in AMBRA1 increases this affinity, we performed isothermal titration calorimetry (ITC) experiments in which synthetic peptides spanning the AMBRA1-LIR at various phosphorylation states (P0: steady state and P1: phosphorylation state) were titrated into all six mATG8 proteins, including LC3A, LC3B, LC3C, GABARAP, GABARAP-L1 and GABARAP-L2 (Supplementary Fig. 4a-c, and Supplementary Note 1). The ITC experiments showed that the unmodified AMBRA1-LIR (P0) binds to LC3B with very low affinity (Fig. 4a left panel). $K_D$ values for this interaction were in the range of >100 μM, a value ~100 higher than the canonical p62-LIR, and comparable to the unphosphorylated NIX- and OPTN-LIR affinities[13,22,23]. However, LC3B affinity to AMBRA1-LIR unambiguously increased upon S1014 phosphorylation, resulting in $K_D$ of 53 μM (Fig. 4a, right panel). Moreover, the nuclear magnetic resonance (NMR) titration experiments indicated a very similar effect of AMBRA1-LIR phosphorylation on its affinity to LC3B (Fig. 4b, c). Interestingly, although ITC experiments showed that the unmodified AMBRA1-LIR (P0) bound all 6 mATG8 proteins with low affinity (Supplementary Fig, 4a, upper plots), it exhibited some preference to GABARAP-subfamily proteins. In fact, strongest interaction appeared to the GABARAP protein, with $K_D$ of ~40 μM, while for the LC3 proteins and for GABARAP-L2 $K_D$ values could be only estimated. Of note, the binding enthalpy for all mATG8 proteins is small, defining the entropy as the main driving force of the interactions. However, both phosphorylation of S1014 (P1, Supplementary Fig. 4b) and introduction of phospho-mimicking aspartate to this position (PM, Supplementary Fig. 4c) increased affinity of AMBRA1-LIR interaction with mATG8 analogues ($K_D$ values decreased ~2–5 times). In particular, phosphorylation of

**Fig. 3** S1014-AMBRA1 phosphorylation is crucial for AMBRA1–LC3 interaction. **a** Sequence alignment of AMBRA1–S1014 site, flanking its LIR motif, in comparison with the S1014A (phospho-dead) and S1014D (phospho-mimetic) generated mutants. **b** HeLa transfected with Myc-AMBRA1[WT], Myc-AMBRA1[S1014A] or Myc-AMBRA1[S1014D] were treated or not with O/A (2.5 μM, 0.8 μM, 1 h) and subjected to co-immunoprecipitation (Co-IP). The graph shows the ratio between Co-IP LC3II/IP-AMBRA1 upon mitophagy induction; $n = 3$. Statistical analysis was performed using Student's $t$-test versus the wild-type form of AMBRA1. **c** PcDNA3, Myc-AMBRA1[WT], Myc-AMBRA1[S1014A] or Myc-AMBRA1[S1014D] transfected HeLa cells were treated with O/A (2.5 μM, 0.8 μM, 4 h) and blotted for COXII, COXIV, Myc and ACTIN; $n = 3$. **d** Cells transfected with a vector encoding for GFP, Myc-AMBRA1[WT], or Myc-AMBRA1[S1014A] or Myc-AMBRA1[S1014D] were treated with O/A as indicated above and stained with anti-Myc and anti-TOM20 (red staining) antibodies. Scale bar, 10 μm; $n = 3$. Myc-AMBRA1[WT] = 9 individual fields; Myc-AMBRA1[S1014A] = 9 individual fields; Myc-AMBRA1[S1014D] = 9 individual fields. **e** Untransfected HeLa cells were treated with mitophagy inducers, such as FCCP (10 μM) and O/A (2.5 μM and 0.8 μM) or with the starvation medium EBSS in order to induce autophagy for 4 h; $n = 3$. Data represent the mean of three different samples (±S.D.) and are representative of experimental triplicate. *$P < 0.05$; **$P < 0.01$; ***$P < 0.001$. Statistical analysis was performed using Student's $t$-test (**b**) or one-way ANOVA (**c, d, e**). $M_r(K)$ = relative molecular mass expressed in kilodalton

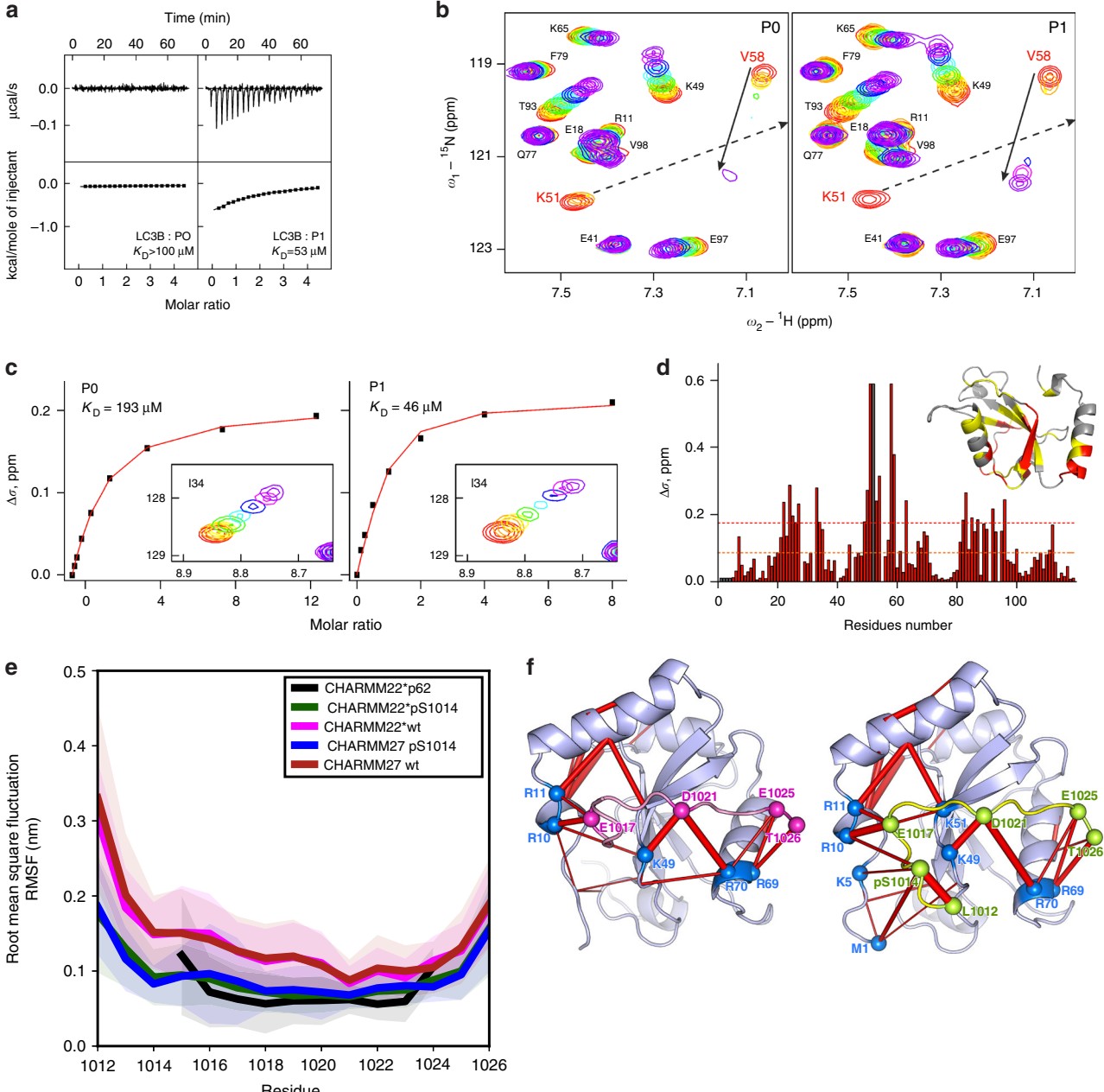

**Fig. 4** S1014-AMBRA1 phosphorylation allows AMBRA1–LC3B binding. **a** ITC titrations of unmodified AMBRA1-LIR peptide (P0) compared with those of S1014-phosphorylated AMBRA1-LIR peptide (P1) into LC3B protein. The measurements were performed at 35 °C. Measured/estimated $K_D$ values are indicated for each interaction. **b** Representative sections of HSQC spectra for ${}^{15}$N-labelled LC3B upon titration with P0 and P1. Both plots show fingerprint regions of the LC3B spectra (around HN resonance of K51). Molar ratios of protein/peptide are rainbow coloured (1:0, 1:0.125, 1:0.25, 1:0.5, 1:1, 1:2, 1:4 and 1:8; from red to violet) for each titration step. Arrows indicate the CSP for K51 and V58 HN backbone resonances. **c** $K_D$ values calculated for the LC3B residue I34 upon titration with P0 and P1. Original CSP values are shown as black squares and the resulting fit is given as a red line in each plot. The original HSQC areas around I34 HN resonance are shown as a box under fitted curves. **d** CSP values ($\Delta\delta$) at the last titration stages for LC3B protein are plotted against residues numbers. The orange dashed line indicates the standard deviations ($\sigma$) over all residues within each dataset, and the red dashed line indicates double $\sigma$ values. The CSP values mapped on the LC3B protein structure (ribbon diagrams, PDB ID 1UGM) are shown in the upper right corner. Residues with small ($\Delta\delta < \sigma$), intermediate ($\sigma < \Delta\delta < 2\sigma$) or strong ($2\sigma < \Delta\delta$) CSPs are marked in grey, yellow and red, respectively. **e** The root mean square fluctuation (RMSF) per residue of AMBRA1 decreases after S1014 phosphorylation (blue, green lines) compared to wt (magenta, red lines) and control P62 (black line) and is persistent with CHARMM22* (black, green and magenta lines) and CHARMM27 (blue and red lines) force fields. Shaded area corresponds to 1 $\sigma$. **f** Superimposition of highly populated (>20%) electrostatic interactions (red bars, thickness indicates occurrence) over the LC3B/AMBRA1-LIR structure. Interactions identified between residues of S1014-phosphorylated AMBRA1-LIR (right panel) and LC3B are identified by spheres and reproduced on the wt (left panel). Similar patterns were seen for simulations using CHARMM22* (shown here) and CHARMM27 force fields

S1014 (P1, Supplementary Fig. 4b, low panel) enhanced the binding with the GABARAP protein and therefore the $K_D$ shifted down to 21 μM, as determined by ITC. Based on chemical shift perturbations (CSP), we precisely calculated $K_D$ values for LC3B interaction with unphosphorylated (P0) and phosphorylated (P1) AMBRA1-LIR peptides. We also mapped the CSP on the LC3B sequence and three-dimensional structure (Fig. 4d). Comparison of the mapped AMBRA1-LIR CSP pattern to previously published mappings indicated that the AMBRA1-LIR interacts with LC3B in a typical LIR mode and phosphorylation on S1014 close to the LIR motif enhances these interactions.

In order to understand the structural impact of S1014-AMBRA1 phosphorylation on the LC3B binding, we performed multi-replicate molecular dynamics (MD) simulations, evaluating conformations of the steady-state structure of AMBRA1 when compared to the phosphorylated structure. In this case, we used the P62-LIR/LC3B experimental structure as a template for modelling. As shown in the plot of Fig. 4e, in analogy to the canonical complex P62-LC3B, the root mean square fluctuation (RMSF), an index of structural flexibility of AMBRA1–LC3B interaction, showed a decrease in flexibility, upon phosphorylation of S1014 (blue and green lines; Fig. 4e). Interestingly, the phospho-variant was characterized by a flexibility profile comparable to the canonical p62-LIR (black line; Fig. 4e), suggesting that this post-translational modification is needed to compensate for the lower binding affinity of the unphosphorylated AMBRA1-LIR, with respect to other known LIR domains. Moreover, we observed that the stabilization induced by the phospho-S1014 was likely due to an increase in electrostatic interactions promoted by this residue (locally and long-range) (Fig. 4f).

Taken together, these results give a proof of concept that phosphorylation of S1014 is fundamental to stabilize the conformation of AMBRA1-LIR in the binding pocket of LC3B, thus promoting its interaction with LC3B (and most likely with all mATG8 proteins) and its mitophagy activity.

**IKKα kinase phosphorylates AMBRA1 at S1014 during mitophagy.** In order to define which kinase could regulate AMBRA1 phosphorylation at S1014 during mitophagy, we performed a bioinformatic analysis by using the Group-based Prediction System (GPS) software[24]. One of the candidates with the highest score was the IKKα kinase (also termed CHUK), a serine–threonine kinase belonging to the IKK family that includes IKKα, IKKβ, IKKγ, IKKε and TBK1 proteins[25]. The predicted IKKα consensus motif[26] was localized upstream of the LIR motif of AMBRA1 (Fig. 5a), suggesting a potential role of IKKα in AMBRA1-mediated mitophagy. To prove this biochemically, we first checked for a putative binding between AMBRA1 and the IKKα kinase. To this end, we over-expressed, in HeLa cells, the kinase-dead mutant of IKKα (HA-IKKα$^{K44M}$)[27] in order to avoid a kiss and run action between the wild-type kinase and AMBRA1, and then we immunoprecipitated the IKKα mutant with an anti-HA antibody and, finally, we checked for AMBRA1–IKKα complex by western blot analysis. As shown in Fig. 5b, we found a binding between the two proteins, suggesting that AMBRA1 could be a substrate for IKKα. To test this hypothesis, we decided to investigate the effect of the well-known IKKα-specific and irreversible inhibitor BAY-117082 on AMBRA1 phosphorylation at S1014 during mitophagy. As shown in Fig. 5c, we observed a clear reduction of AMBRA1 phosphorylation on S1014, when cells were treated with BAY-117082 compared to cells treated with the vehicle (dimethyl sulfoxide). These results suggested that the IKKα kinase was responsible for the phosphorylation on S1014 of AMBRA1 following mitophagy induction. To strengthen

these data, we decided to test the effect of the kinase-dead form of IKKα (IKKα$^{K44M}$), which acts as a dominant negative mutant[27], on AMBRA1 phosphorylation at S1014. To this end, we over-expressed vectors encoding Flag-IKKα$^{WT}$ or HA-IKKα$^{K44M}$ in HeLa cells, and we induced mitophagy by using O/A. As shown in Fig. 5d, the expression of the kinase-dead version of IKKα was sufficient to abolish AMBRA1 phosphorylation at S1014 following O/A treatment, similarly to BAY-117082 (see Fig. 5c). Consistent with these results, we validated the IKKα-mediated phosphorylation on AMBRA1 at S1014 by an in vitro kinase assay, comparing the activity of IKKα$^{WT}$ to its kinase-dead mutant. To this end, we transfected HeLa cells with a vector encoding Myc-AMBRA1$^{WT}$, Flag-IKKα$^{WT}$ or HA-IKKα$^{K44M}$ and immunoprecipitated individually these three proteins from the mitochondrial fraction. Then, we combined the immunoprecipitated kinases (IP-IKKα$^{WT}$ or IP-IKKα$^{K44M}$) with the immunoprecipitated substrate (IP-AMBRA1). The western blot analysis showed that the phosphorylation signal at AMBRA1–S1014 was present only in the IP-IKKα fraction (Fig. 5e). Subsequently, we also performed an in vitro kinase assay with human recombinant AMBRA1 as a substrate in order to formally prove a direct effect of the IKKα kinase on its novel substrate. As shown in Fig. 5f, the phosphorylated form of AMBRA1 at S1014 was detected when mixed with the IP-IKKα, but was not appreciated in the kinase-dead IP-IKKα$^{K44M}$ sample. Despite TBK1 and IKKα kinases shared a similar structure, we did not observe any effects of the TBK1 kinase on S1014 phosphorylation of AMBRA1 (Supplementary Fig. 5a).

These results prove that IKKα is the kinase responsible for AMBRA1 phosphorylation at S1014 upon mitophagy induction, and thus it represents a novel upstream factor accountable for mitophagy activation.

**IKKα inhibition impairs AMBRA1-mediated mitophagy.** In order to verify whether the abolishment of IKKα kinase activity could inhibit AMBRA1-mediated mitophagy, we tested the effect of IKKα inhibition on AMBRA1-mediated mitophagy. To this end, we checked for mitophagy induction in HeLa cells over-expressing AMBRA1$^{ActA}$, and treated or not with the IKKα inhibitor (BAY-117082) by detecting COXII and COXIV levels by western blot analysis. We observed that BAY-117082 treatment was able to rescue mitochondrial clearance following AMBRA1$^{ActA}$ overexpression (Fig. 5g and Supplementary Fig. 5b).

Of the highest importance, the IKKα inhibitor also impacted AMBRA1-mediated mitophagy in Myc-AMBRA1$^{WT}$ transiently transfected HeLa cells, in which we induced mitophagy by O/A. As shown in Fig. 5h and Supplementary Figure 5c, by analysing the mitochondrial clearance by western blotting, we also observed in this case a rescue of the mitochondrial markers COXII and COXIV upon BAY-117082 treatment.

To validate these data, we co-transfected HeLa cells with a vector encoding Myc-AMBRA1$^{ActA}$ in combination with HA-IKKα$^{K44M}$ kinase-dead mutant. Again, by blocking IKKα activity, by using the dominant negative form of the kinase, we induced AMBRA1$^{ActA}$-mediated mitophagy inhibition (Fig. 5i and Supplementary Fig. 5d). As expected, AMBRA1$^{WT}$-induced mitophagy was also reduced in HA-IKKα$^{K44M}$ transfected cells, confirming that inhibition of IKKα also impaired AMBRA1$^{WT}$-mediated mitophagy (Fig. 5j and Supplementary Fig. 5e). Moreover, as a control, we checked whether the nuclear factor (NF)-κB pathway was involved in this regulation by looking at p65 nuclear translocation upon mitophagy stimulation. As shown in Supplementary Figure 5f, no evident p65 translocation to the nucleus was observed upon O/A treatment compared to tumour necrosis

factorα treatment, indicating that this inflammation pathway was not activated during mitophagy.

Altogether, these results point to IKKα as an essential factor in AMBRA1-mediated mitophagy, regulating AMBRA1 phosphorylation at S1014, and highlight a novel and alternative role (rather than in inflammation) of IKKα in mitophagy.

**HUWE1 controls the S1014 phosphorylation of AMBRA1.** In analogy with other mitophagy receptors, the phosphorylation of AMBRA1 at the S1014 site, flanking its LIR motif, is a crucial event for the engulfment of ubiquitin-tagged mitochondria into autophagosomes. Since we demonstrated that HUWE1 down-regulation results in a block of AMBRA1-mediated mitophagy,

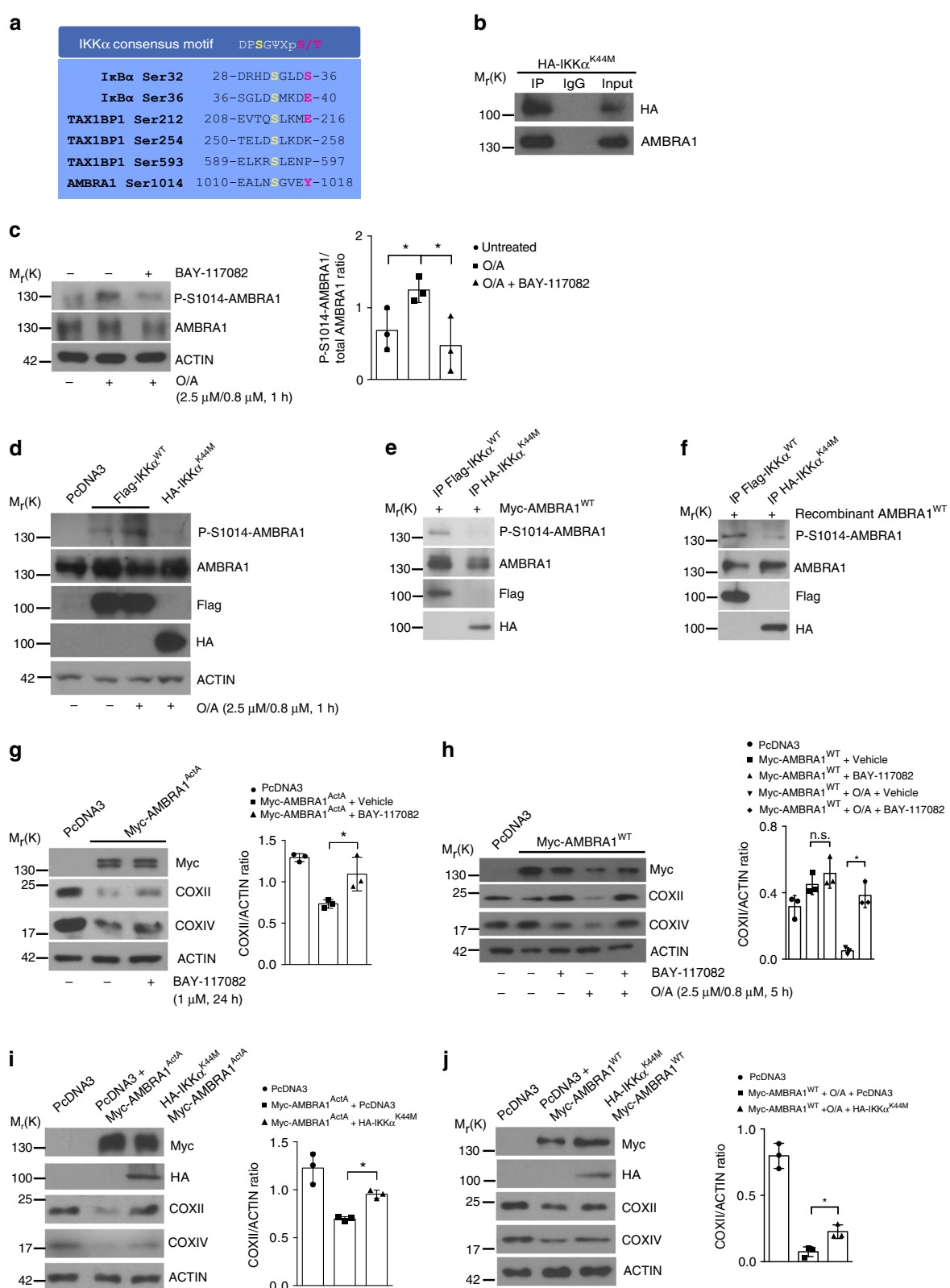

**Fig. 5** IKKα controls AMBRA1 phosphorylation at S1014 during mitophagy. **a** Sequence alignment of the consensus sequence of IKKα-phosphorylation on IκBα, TAX1BP1 and AMBRA1. **b** HA-IKKα$^{K44M}$ transfected HeLa cells were subjected to Co-IP and blotted for the indicated antibodies; $n = 3$. **c** Untransfected HeLa cells were treated with O/A (2.5 µM, 0.8 µM, 1 h) and with the IKKα irreversible inhibitor BAY-117082 (2 µM, 1 h); $n = 3$. **d** Representative image of HeLa cells transfected with PcDNA3, Flag-IKKα$^{WT}$ or the kinase-dead HA-IKKα$^{K44M}$, treated with O/A and blotted for the indicated antibodies; $n = 3$. **e** In vitro kinase assay of immunoprecipitated-Flag-IKKα$^{WT}$- or HA-IKKα$^{K44M}$-expressing HeLa cells in which the substrate (Myc-AMBRA1$^{WT}$) was transcribed/translated in vitro. Samples were immunoblotted for the indicated proteins; $n = 2$. **f** In vitro kinase assay in which the immunoprecipitated fraction of IKKα$^{WT}$ or IKKα$^{K44M}$ were mixed to recombinant produced AMBRA1; $n = 2$. **g** HeLa cells transfected with Myc-AMBRA1$^{ActA}$ or PcDNA3 (24 h) and treated with BAY-117082 inhibitor were blotted for the indicated antibodies; $n = 3$. **h** HeLa cells transfected with Myc-AMBRA1$^{WT}$ or PcDNA3 and treated with O/A (2.5 µM, 0.8 µM, 5 h) and BAY-117082 inhibitor (2 µM, 5 h) were immunoblotted for the indicated proteins; $n = 3$. **i** HeLa cells transfected with PcDNA3 or HA-IKKα$^{K44M}$ constructs in combination with Myc-AMBRA1$^{ActA}$ were analysed by western blot for the indicated antibodies; $n = 3$. **j** PcDNA3+Myc-AMBRA1$^{WT}$ or Myc-AMBRA1$^{WT}$+HA-IKKα$^{K44M}$ co-transfected HeLa cells treated with O/A were subjected to immunoblotting for COXII, COXIV, HA, Myc and ACTIN antibodies; $n = 3$. Data represent the mean of three different samples (±S.D.) and are representative of experimental triplicate. *$P < 0.05$. Statistical analysis was performed using one-way ANOVA (**c, g, h, I, j**). M$_r$(K) = relative molecular mass expressed in kilodalton

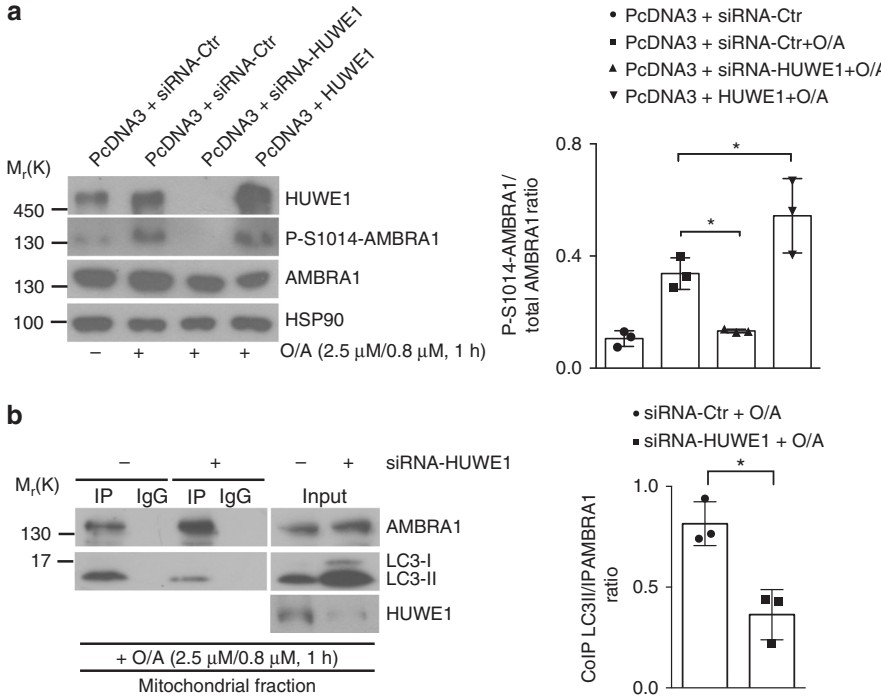

**Fig. 6** HUWE1 is crucial for AMBRA1–S1014 phosphorylation. **a** HeLa cells were transfected with an empty vector (PcDNA3) in combination with a siRNA-Ctr or a siRNA-HUWE1 or a plasmid encoding for HUWE1. Cells were treated with O/A (2.5 µM, 0.8 µM, 1 h) and total lysates were blotted for the indicated antibodies; $n = 3$. **b** HeLa cells were transfected with siRNA-Ctr or siRNa-HUWE1 and treated with O/A (2.5 µM, 0.8 µM, 1 h). Mitochondrial fractions were immunoprecipitated using anti-AMBRA1 antibody; $n = 3$. Data represent the mean of three different samples (±S.D.) and are representative of experimental triplicate. *$P < 0.05$. Statistical analysis was performed using one-way anOVA (**a**) or Student's $t$-test (**b**). M$_r$(K) = relative molecular mass expressed in kilodalton

we hypothesized that HUWE1 activity could influence AMBRA1 phosphorylation at S1014 upon mitophagy induction. Thus, we transfected HeLa cells with a siRNA-Ctr or siRNA-HUWE1 or with a vector encoding HUWE1, and then treated cells with O/A (2.5 µM, 0.8 µM, 1 h).

Interestingly, we observed that the downregulation of HUWE1 induced a significant reduction in AMBRA1–S1014 phosphorylation upon mitophagy induction. By contrast, HUWE1 overexpression resulted in increased phosphorylation of AMBRA1 at the same site (Fig. 6a). In analogy with this result, we observed that HUWE1 depletion produces a reduction in AMBRA1–LC3B binding upon mitophagy stimulation (Fig. 6b).

Taken together, these findings clearly define an epistatic role of HUWE1 in IKKα- and AMBRA1-mediated mitophagy.

**AMBRA1 regulates mitochondrial clearance during ischaemia.** Following the evidence on AMBRA1-mediated mitophagy in relay with HUWE1 and IKKα, we checked the physiopathological relevance of this functional interplay in models of ischaemia. Indeed, there is general consensus that mitophagy can be triggered in similar conditions as it happens via receptors. For instance, NIX/BNIP3L mediates mitochondrial removal during cerebral[28] and cardiac[29] ischaemia/reperfusion (I/R) injury; also, FUNDC1 is a well-known regulator of mitophagy under hypoxia[11], and its implications in ischaemia have been recently elucidated[30,31]. Of note, the intramitochondrial protein ATPIF1, during ischaemia-preconditioning context, has been reported to regulate selective degradation of potentially toxic mitochondria (i.e., those generating high doses of free radicals) by collapsing their membrane potential (ΔΨm)[32,33].

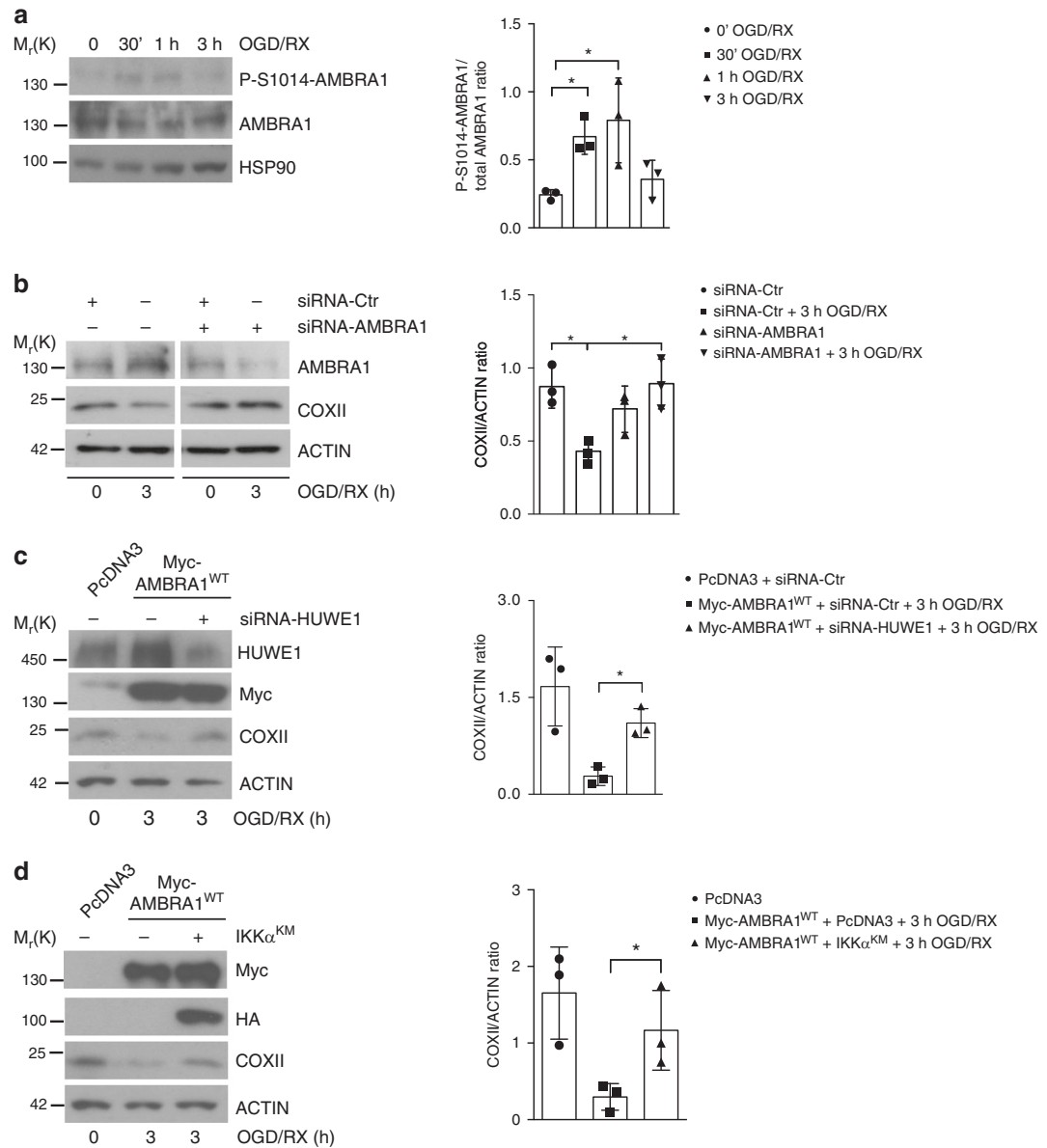

**Fig. 7** AMBRA1 induce mitophagy in an in vitro model of ischaemia. **a** Lysates of SH-SY5Y cells exposed to different time points of OGD/RX (0 min, 30 min, 1 h and 3 h) were blotted with the indicated antibodies; $n = 3$. **b** Endogenous AMBRA1 levels were downregulated using siRNA-AMBRA1 in SH-SY5Y cells. Following exposure to 3 h OGD/RX, total lysates were subjected to western blot analysis in order to analyse mitochondria clearance, looking at COXII protein levels; $n = 3$. **c** PcDNA3 or Myc-AMBRA1$^{WT}$-expressing SH-SY5Y cells were transfected with siRNA-Ctr or siRNA-HUWE1 constructs and then immunoblotted for the indicated proteins; $n = 3$. **d** SH-SY5Y cells transfected with PcDNA3 or HA-IKKα$^{K44M}$ constructs in combination with Myc-AMBRA1$^{WT}$ were subjected to western blot analysis for the indicated antibodies; $n = 3$. Data represent the mean of three different experiments (±S.D.). *$P < 0.05$. Statistical analysis was performed using one-way ANOVA. $M_r(K)$ = relative molecular mass expressed in kilodalton

Ischaemia and its prodromal phases are characterized by detrimental oxidative stress, which we showed to be counteracted by AMBRA1-induced mitophagy[34]; this evidence prompted us to investigate whether AMBRA1 itself could drive mitophagy during I/R injury.

We thus tested the degree of AMBRA1 phosphorylation in SH-SY5Y cells exposed to an in vitro model of ischaemia mimicked by alternative time periods of oxygen-glucose deprivation/re-oxygenation (OGD/RX)[33]. Total lysates were then subjected to western blot analysis of P-S1014-AMBRA1, recording that AMBRA1 is neatly phosphorylated between 30 min and 1 h of OGD/RX treatment (Fig. 7a). Nevertheless, the role of endogenous AMBRA1 in this physiopathological activation of mitophagy was further corroborated by downregulating AMBRA1 in SH-

SY5Y cells exposed to the same OGD/RX protocols. In Fig. 7b, the decrease of COXII is utterly rescued when AMBRA1 is removed. Subsequently, we checked whether HUWE1 and IKKα also actively took part to AMBRA1-mediated mitophagy upon OGD/RX. In Myc-AMBRA1$^{WT}$-expressing SH-SY5Y cells, HUWE1 expression was repressed via siRNA; this primed a significant rescue of COXII during OGD/RX (Fig. 7c). In another experiment, we also abrogated IKKα activity by overexpressing the dominant negative kinase IKKα$^{K44M}$ in SH-SY5Y cells transfected with Myc-AMBRA1$^{WT}$. Indeed, 3 h of exposure to OGD/RX impaired the clearance of mitochondria (Fig. 7d). In sum, these data lead us to conclude that AMBRA1, together with HUWE1 and IKKα, may well represent a novel regulator of ischaemia-induced mitophagy, thus highlighting the

physiopathological relevance of this pathway in mitochondria quality control.

## Discussion

In the present study, we demonstrated that the mitophagy receptor AMBRA1 acts at two critical steps during mitochondrial membrane depolarization-induced mitophagy. First, we identified the E3 ubiquitin ligase HUWE1 as an important co-operator of AMBRA1 in order to induce mitophagy, following mitochondrial membrane depolarization. In particular, we demonstrated that AMBRA1[ActA] strongly favours the interaction between HUWE1 and MFN2, thus acting as a cofactor for HUWE1 E3 ubiquitin ligase activity, and leading subsequently to MFN2 ubiquitylation and then to its degradation. The data highlight a novel important paradigm for HUWE1 E3 ubiquitin ligase activity regulation and open a wide range of studies to understand whether AMBRA1 may control HUWE1 activity in other contexts. Of note, HUWE1 depletion abolishes both MFN2 degradation and AMBRA1-mediated mitophagy in HeLa cells. Also, it has been established that MFN2 degradation is an important event for mitophagy induction[35]. In line with this evidence, we showed that MFN2 is degraded in the early steps of AMBRA1-mediated mitophagy (18 h post transfection of Myc-AMBRA1[ActA]). Moreover, HUWE1 depletion induces a clear reduction of total mitochondria ubiquitylation following AMBRA1[ActA] expression. Since we found that MFN2 degradation correlates with AMBRA1-mediated mitophagy induction, it will be necessary to better understand in the future whether or not this event is a critical step for AMBRA1-mediated mitophagy. Also, although MNF2 is a favourite substrate of HUWE1, it will be important to investigate whether HUWE1 could ubiquitylate other mitochondrial substrates in order to control mitophagy induction.

We recently demonstrated that AMBRA1 is a BH3 (BCL2-homology domain 3)-like protein, containing a BH3 domain necessary for its interaction with BCL2 family proteins[36,37]. Interestingly, HUWE1 has been defined as a unique BH3-containing E3 ubiquitin ligase[38]; it would be interesting to characterize the binding between AMBRA1 and HUWE1 by checking, in particular, whether a BH3 mechanism is involved in this context. Of note, both HUWE1 and AMBRA1 can interact with the anti-apoptotic factor MCL1 (induced myeloid leukaemia cell differentiation protein Mcl-1)[36,38]. In this scenario, it could also be interesting to define the relationship among AMBRA1–HUWE1-dependent mitophagy, mitochondrial dynamics and apoptosis.

The HUWE1–AMBRA1 axis in mitophagy defines a new pathway for damaged mitochondrial removal related to a novel E3 ligase, yet unwound in this field, but since AMBRA1 and HUWE1 are two large multi-domain proteins, they may cooperate in several processes, such as cancer, apoptosis and cell proliferation. In fact, HUWE1 has been described as a tumour suppressor gene controlling the proto-oncogene c-Myc regulation. HUWE1 participates to c-Myc degradation, leading to a reduction in tumour diffusion[39]. It is opportune to underline that AMBRA1 also controls c-Myc expression, cooperating in the PP2A-mediated dephosphorylation of c-Myc[40]. It could thus be interesting to analyse the correlation between AMBRA1–HUWE1 and c-Myc regulation upon mitophagy stimulation. As previously demonstrated, mitochondria are fundamental in cellular homoeostasis and the HUWE1–AMBRA1 axis can represent a focal point of interconnection between healthy and tumour cells.

In addition, it is noteworthy that MFN2 degradation is also mediated by PARKIN activity[41]. Since we demonstrated that AMBRA1-dependent mitophagy is not controlled by the PINK1/PARKIN pathway, we propose here an alternative system that regulates mitophagy. The removal and proteolysis of OMM proteins appears to be necessary for mitophagy[42]. Since the PINK1/PARKIN pathway is mutated in one of the most diffuse neurodegenerative disturb, Parkinson's Disease[43,44], AMBRA1–HUWE1-dependent mitophagy could offer a novel target therapy to counteract this disease as it may preserve cellular vitality. The mitophagy programme is a highly intricate system comprising different characters. In fact, PINK1-related mitophagy receptors present UBDs in their sequences that allow the interaction with Ser65-phosphoUb-containing mitochondria[5,45]. By contrast, mitochondrial proteins, such as NIX, FUNDC1 and Bcl2-L-13, lacking an UBD, can anyway promote mitophagy of ubiquitylated mitochondria. In our current work, we added a piece to this heterogeneous puzzle, describing AMBRA1 as a new UBD-free mitophagy receptor, which acts following mitochondrial membrane depolarization. Indeed, we revealed that the expression of AMBRA1 is sufficient to restore mitochondrial clearance in cells depleted for PINK1/PARKIN-related mitophagy receptors, such as NDP52, OPTN, TAX1BP1, P62 and NBR1. In analogy with mitophagy receptors Bcl2-L-13, FUNDC1 and NIX/BNIP3L, we demonstrated that AMBRA1 is also able to stimulate mitochondrial clearance in the absence of any PINK1 phospho-ubiquitin signals. Indeed, we previously proved that AMBRA1 was able to induce mitophagy in PINK1[−/−] MEFs cells[13].

Moreover, we have proved that AMBRA1-mediated mitophagy activity is tightly dependent on its phosphorylation state. In particular, following mitophagy induction, the phosphorylation on S1014 increases the AMBRA1–LC3B binding thus promoting an enhancement in AMBRA1-mediated mitophagy. Moreover, we have revealed that the HUWE1 ubiquitylation signal is fundamental not only in tagging mitochondria for mitophagy, but also in activating AMBRA1 for mitophagy. In fact, we observed that downregulation of HUWE1 results in a reduction of AMBRA1-activated phosphorylation state, necessary for its interaction with the autophagosome marker LC3B. We can speculate that a fine dual relationship occurs between AMBRA1 and HUWE1: the former is necessary for HUWE1 recruitment to the mitochondria; the latter, once at the mitochondria, ubiquitylates OMM protein(s), thus inducing mitophagy and producing a signal for AMBRA1–S1014 phosphorylation, flanking its LIR motif. This event causes in turn a structural rearrangement necessary to make the AMBRA1-LIR motif more accessible to LC3B. In particular, we detected that the negative charge on S1014 of AMBRA1 produces an increase in electrostatic interactions in the proximity of the tryptophan site (W1020) on the AMBRA1-LIR motif. As shown in Supplementary Fig. 4, the importance of electrostatic interactions involving side chains of mATG8 proteins is confirmed by the NMR titration data. By performing ITC and NMR experiments, we observed that the phosphorylation on S1014 is important to increase the affinity between AMBRA1-LIR and mATG8 proteins (Supplementary Fig. 4) and this result led us to identify GABARAP proteins as more specific interacting proteins of AMBRA1. Of note, the S1014 site is not directly located prior to the core LIR aromatic residue W1020 of AMBRA1. Therefore, the increase of AMBRA1–mATG8 binding by S1014 phosphorylation is a quite unexpected event. For example, the OPTN phosphorylation on S170, flanking its LIR motif, shows a small increase in affinity to LC3B[24]. Based on this, we cannot exclude that other factors (such as additional residues or structural elements in AMBRA1, or AMBRA1 additional post-translational modifications) could also affect the interactions between full-length AMBRA1 and mATG8 proteins. Moreover, since GABARAPs seem to be more involved in the PINK1/PARKIN-dependent mitophagy pathway than LC3s[46], it will be very interesting to investigate, in the future, whether AMBRA1 requires a preferentially binding to

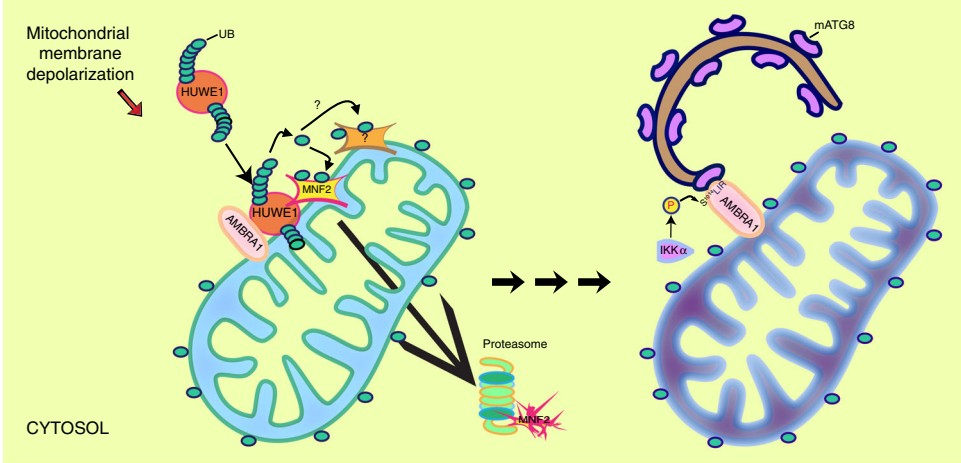

**Fig. 8** AMBRA1 regulates mitophagy at two critical steps. Upon mitophagy stimulation, AMBRA1 mediates the HUWE1 E3 ubiquitin ligase translocation from cytosol to mitochondria (light blue). AMBRA1 acts as a cofactor for HUWE1 E3 ubiquitin ligase activity, favouring its binding to its substrate MFN2 (and maybe other OMM substrates) and targeting it to the proteasome. This event is crucial and required for AMBRA1-induced mitophagy. In a second step, in analogy with other mitophagy receptors, AMBRA1 is phosphorylated at S1014, flanking its LIR motif, in order to be able to interact with mATG8s. The final outcome is the engulfment of damaged mitochondria (dark blue) into autophagosomes. This post-translational modification is controlled by IKKα kinase upon mitophagy stimulation. Reproduced from ref.[71] with permission from Elsevier

GABARAPs or to LC3s in mitophagy, depending on a particular context (PINK1/PARKIN dependent or independent) or on other processes (i.e., macroautophagy or different selective forms of autophagy).

Our results clearly define the importance of post-translational modifications on the phosphorylated site, in proximity of the identified LIR motif of AMBRA1. Several reports demonstrated the central role of kinase-mediated phosphorylation in cargo receptor-mediated autophagy[11,47]. We here demonstrated a novel role of IKKα, independently of its role in NF-κB pathway in mitophagy. Interestingly, Criollo et al.[48] proved that the IKK complex was able to promote macroautophagy activation in an NF-κB-independent manner.

Although the IKK family is composed of several members, including IKKα, IKKβ, IKKγ, TBK1 and IKKε, the structure of these factors is very similar[25]. Interestingly, one of these proteins, TBK1, plays a pivotal role in controlling selective autophagy, such as xenophagy and mitophagy[6,23,47,49]. Here, we demonstrated that, in analogy with TBK1, which is mainly described to be involved in the phosphorylation of the mitophagy receptor OPTN upstream of its LIR motif in xenophagy[23] and in mitophagy[5], IKKα is able to phosphorylate the mitophagy receptor AMBRA1 flanking its LC3-interacting region during mitochondrial clearance. It seems that such mechanisms are evolutionary conserved and we hypothesize that similar to OPTN, AMBRA1 could also be involved in other selective autophagy processes (i.e., xenophagy).

It is noteworthy that IKKα is also fundamental in mitochondrial dynamics regulation by favouring OPA1 activation and therefore mitochondrial fusion[50]. Moreover, Bakkar et al.[51] demonstrated that IKKα regulates mitochondrial biogenesis in muscle tissue. These data underline the complexity of IKKα function, which seems to be highly related to inflammation, mitochondria dynamics and mitophagy.

In sum, we prove that previously unravelled mitophagy-involved proteins, HUWE1 E3 ubiquitin ligase and IKKα kinase, play pivotal roles in AMBRA1-mediated mitophagy. HUWE1 is crucial to trigger AMBRA1-mediated mitophagy by favouring MFN2 degradation. This event is necessary for AMBRA1 phosphorylation on its S1014-mediated by IKKα. This kinase, in turn,

regulates the engulfment of damaged mitochondria into autophagosomes by phosphorylating AMBRA1–S1014, thus enabling AMBRA1–mATG8 interactions. Since either HUWE1 or IKKα depletion inhibit mitophagy mediated by AMBRA1, we propose that these two proteins strongly contribute to two different steps in the AMBRA1-mediated mitophagy control (Fig. 8).

Finally, we demonstrated that AMBRA1, in a model of ischaemia/re-oxygenation, regulates mitophagy together with HUWE1 and IKKα, thus implying that AMBRA1-mediated mitophagy plays a protective role in response to hypoxia. In the same line, it would also be interesting to test whether *Ambra1* heterozygous mice fail to show protection against ischaemic stress.

In sum, this work highlights this novel and alternative mitophagy pathway as a new target in developing therapy to counteract diseases characterized by malfunctioning mitochondria such as neurodegeneration, hypoxic-ischaemic encephalopathy and cancer.

## Methods

**Antibodies**. Antibodies used for western blot or immunofluorescence analysis are listed in the Supplementary Table 1 with the appropriated dilutions.

**Cell cultures and transfection**. Cells were cultured in Dulbecco's modified Eagle's medium (DMEM; GIBCO 41966-029) supplemented with 10% foetal bovine serum (FBS; Thermo Fisher Scientific, 10270-106) and 1% penicillin–streptomycin solution (Lonza, 17-602 E) at 37 °C under 5% $CO_2$. HeLa and SH-SY5Y cells were purchased from Sigma-Aldrich (93021013 and 9403304, respectively). Penta-knockout HeLa cells (Penta-KO) and mt-mKeima expressing wild-type or PINK1 knockout HeLa cells (mt-mKeima HeLa; mt-mKeima PINK1-KO) were a gift of R. Youle[5]. HEK293T-HUWE1 cells were donated by T. Mak.

Transient transfections of expression plasmids into HeLa, Penta-KO, HEK293T-HUWE1 and SH-SY5Y cells were performed using TurboFect or Lipofectamine 2000 according to the supplier's instructions (Thermo Fisher Scientific, R0532; Invitrogene, 11668019).

**Plasmids and cloning**. All the transfected constructs presented in this article are described in Supplementary Table 2.

Mutations listed in the Supplementary Table 3 were generated using the QuickChange site-directed mutagenesis kit (Stratagene, 200519) and all plasmids generated in this study were verified by DNA sequencing (Eurofins).

**SiRNA-mediated knockdown**. RNA-interference was performed using Lipofectamine 2000. A list of siRNA constructs is described in Supplementary Table 4.

**Cell treatment**. After 6 h of transfection or 24 h after cell seeding, cells were treated as described below. Penta-KO were treated with the combined treatment O/A (10 μM oligomycin (Calbiochem)/4 μM antimycin A (Sigma)), supplemented with 20 μM QVD (Sigma-Aldrich, SML0063) in fresh growth medium for 24 h, as in ref. [5]. HeLa cells were treated either with the combined treatment O/A (2.5 and 0.8 μM) or FCCP 10 μM (Sigma-Aldrich, C2920) or EBSS (Thermo Fisher Scientific, 24010043) for the indicated time. In order to block IKKα activity, cells were treated with the irreversible inhibitor BAY-117082 at 1 or 2 μM, for the indicated time. Autophagosome–lysosome fusion was blocked with NH$_4$Cl 40 mM (Sigma-Aldrich, O9718).

**Western blot analysis**. Cells were rinsed in phosphate-buffered saline (PBS) on ice and lysed in RIPA buffer supplemented with a protease inhibitor cocktail (Sigma-Aldrich, P8340), Na$_4$VO$_3$ 0.1 mM (Sigma-Aldrich, S6508), NaF 1 mM (Sigma-Aldrich, S7920) and β-Glycerophosphate 5 mM (Sigma-Aldrich, G6376).

Cell extracts were centrifuged at 15,000 × g for 10 min at 4 °C. Protein concentrations were determined with the Bio-Rad Protein Assay Kit (Bio-Rad, 5000001). Cell extracts or immunoprecipitates were separated by sodium dodecyl sulphate–polyacrylamide gel electrophoresis (SDS–PAGE). Polyvinylidene difluoride (PVDF; Merck-Millipore, IPVH00010) or Nitrocellulose (GE Healthcare, 10600002) membranes were incubated with primary antibodies followed by horseradish peroxidase-conjugate secondary antibodies (Bio-Rad, 1706515 and 1721011) and visualized with ECL (Merck-Millipore WBKLS0500). Uncropped scans of the most important blots are reported in the Supplementary Information, as Supplementary Figure 6.

**Co-immunoprecipitation (Co-IP)**. After cell lysis, equal amounts of protein were incubated with primary antibodies with end-over-end rotation for 24 h. Then, 30 μl of protein A agarose beads (Roche, 11719408001) were added for 60 min. The beads were collected by centrifugation and washed 3 times with the RIPA buffer. Immunocomplexes were eluted with 30 μl of SDS sample buffer and heated at 95 °C for 10 min.

**Immunofluorescence analysis**. Cells were washed in PBS and fixed with 4% paraformaldehyde in PBS for 10 min. After permeabilization with 0.4% Triton X-100 (Sigma-Aldrich, X-100) in PBS for 5 min, cells were incubated overnight at 4 °C with primary antibodies and 2% normal goat serum (Sigma-Aldrich, G9023). Cells were then washed with PBS (GIBCO, BE17-512F) and incubated for 1 h with labelled anti-mouse (Thermo Fisher Scientific, A11017-A21425) or anti-rabbit (Thermo Fisher Scientific, A11070-A21430-A31573) secondary antibodies. Nuclei were stained with 1 μg/ml DAPI (4′,6-diamidino-2-phenylindole) and examined under a Zeiss LSM 700 100× oil-immersion objective (CLSM700; Jena, Germany). We used ImageJ software for image analysis. We calculated the mito content as percentage of cytosolic area occupied by mitochondria with Mitophagy macro[52]. For Supplementary Figure 1b, the quantification as mito-aggresome-positive transfected cells (%) was performed by a blind approach (the information on plasmid transfection was masked), counting the mitochondria with round and clumped shape. All acquisitions were performed in non-saturated single z-confocal planes.

**Nuclei–mitochondria–cytosol purification**. Cells, rinsed in ice-cold PBS and centrifuged at 800 × g for 10 min, were resuspended in homogenization buffer (HB: 0.25 M sucrose (Sigma-Aldrich, S0389), 10 mM HEPES pH 7 (Sigma-Aldrich, H4034), EDTA (Sigma-Aldrich, ED-100)) with protease and phosphatase inhibitors cocktail. Nuclei, mitochondria and cytosol purifications proceed by standard differential centrifugations. After homogenization of the solution with 30–70 pulses (one pulse corresponds to approximately 1 s), cells were collected at 600 × g at 4 °C for 10 min. The pellet containing the nuclear fraction was suspended in RIPA buffer, instead the supernatant was subjected to a further centrifugation at 11,000 × g for 15 min at 4 °C. The mitochondrial pellet fraction was suspended in RIPA buffer. The supernatant fraction corresponded to soluble cytosolic proteins.

**In vitro IP-kinase assay**. HeLa cells were transiently transfected with a vector encoding for Myc-AMBRA1$^{WT}$ or Flag-IKKα$^{WT}$ or HA-IKKα$^{K44M}$. After 24 h, equal amounts of proteins (500 μg) were immunoprecipitated using anti-Myc, anti-Flag and anti-HA antibodies, respectively, overnight. Protein A Agarose beads (30 μl of 50% slurry) (Roche, 11719408001) were added to the samples at 4 °C with end-over-end rotation for 1 h. The immunoprecipitates were collected by centrifugation and washed 3 times with RIPA buffer. Pellets were washed for 3 min with 500 μl 1× Kinase Buffer (KB: 20 mM Hepes pH 7.4, 10 mM MgCl$_2$, 25 mM Glycerophosphate, 0.1 Na$_4$VO$_3$, 4 mM NaF, 1 mM DTT (Sigma-Aldrich, GE17-1318-01)). The IP-Flag- IKKα$^{WT}$ or IP-HA-IKKα$^{K44M}$ final pellets were resuspended in 50 μl of 1× Kinase Buffer, while IP-Myc-AMBRA1 was suspended in 50 μl of RIPA buffer. In order to test kinase activity with a non-radioactive method, 1 μl of 10 mM adenosine triphosphate (ATP) and 20 μl of IP-Myc-AMBRA1 substrate were added to the IP kinases for 30 min at 30 °C. The reaction was stopped by adding 30 μl of 4× SDS Sample Buffer and denaturizing at 95 °C for 5 min. For

the kinase assay with recombinant AMBRA1, the wild-type AMBRA1 protein was transcribed and translated in vitro with standard procedures (Promega, L1170 TnT® Quick Coupled Transcription/ Translation Systems). At the end of IKKα$^{WT}$ or IKKα$^{K44M}$ immunoprecipitation (as described above), 1 μl of 10 mM ATP and 2 μl of kinase substrate was added to the IP-kinases for 30 min at 30 °C. The reaction was stopped by adding 30 μl of SDS Sample Buffer and denaturizing at 95 °C for 5 min. For both of these kinase assays, samples were analysed by western blot.

**Isothermal titration microcalorimetry**. The ITC experiments were performed at 25 °C and 35 °C using a MicroCal VP-ITC microcalorimeter (Malvern Instruments Ltd, UK). The peptides at concentrations of 0.35 mM were titrated into 0.015 mM LC3 and GABARAP proteins in 21 steps. The ITC data were analysed with the ITC-Origin 7.0 software with a one-site binding model. The protein and peptide concentrations were calculated from the UV absorption at 280 nm through a Nanodrop spectrophotometer (Thermo Fisher Scientific, DE, USA).

**NMR spectroscopy**. All NMR experiments were performed at 298 K on Bruker Avance spectrometer operating at a proton frequencies of 600 MHz. Titration experiments were performed with 0.17 mM $^{15}$N-labelled LC3B and GABARAP protein samples to which the non-labelled AMBRA1-LIR peptides were added stepwise until 8 times excess to LC3B protein, or 4 times excess to the GABARAP protein. Chemical shift perturbations (CSP) analysis was done according to the most recent review[53]. CSP values were calculated for each individual backbone proton using the formula: $\Delta\delta = [\Delta\delta_H{}^2 + (\Delta\delta_N/5)^2]^{1/2}$. The dissociation constants, $K_D$, were calculated by a least squares fit to the titration data under the assumptions of a fast exchange regime and a one binding site mode of the protein interaction. The obtained CSP values were mapped on LC3B (PDB ID 1UGM[54]) and GABARAP (PDB ID 1GNU[55]) structures.

**ITC and NMR sample preparation**. For ITC and NMR experiments, the unlabelled or $^{15}$N-labelled LC3 and GABARAP proteins were obtained based on the protocols described elsewhere[56,57]. AMBRA1-LIR-containing peptides with different states of phosphorylation (unphosphorylated, P0: EALNSG-VEYYWDQLNET; D-phosphomimicked, PM: EALNDGVEYYWDQLNET; and S1014 phosphorylated, P1: EALN{pS}GVEYYWDQLNET) were purchased from GenScript Inc. (NJ, USA). Before experiments, all proteins and peptides were equilibrated with a buffer containing 50 mM Na$_2$HPO$_4$, 100 mM NaCl, pH 7.0; and supplemented with 5 mM protease inhibitor cocktail.

**AMBRA1–LC3B modelling**. We used Modeller version 9.15[58] to generate structural models of the AMBRA1-LIR motif (L$_{1013}$NSGVEYYWDQLNET$_{1026}$) in complex with LC3B to use as starting structures for the all-atom MD simulations. For the modelling, we used the X-ray structure of p62-LIR motif in complex with LC3B (PDB entry 2ZJD[59]) as a template. We generated 1000 binding poses with Modeller and we then carried out structural clustering with the linkage algorithm based on the root mean square deviation (RMSD) of all the atoms of Ambra1-LIR peptide. We selected the average structures from each of the four more populated clusters (i.e., the conformer with the lowest RMSD to the other ones in the same cluster) and we used each of them as a starting structure for all-atom explicit solvent molecular simulations using multiple replicates. For the phosphorylated variant of Ambra1-LIR, we generated a second set of models by in silico phosphorylation (S2P; phosphoserine with a −2 charge) on position S1014 in each of the four models previously selected using the web interface of Vienna-PTM[60]. We treated in our simulation the phosphoserine S1014 using the −2 state. Indeed, the pKa values of single protonated/un-protonated transition of phosphoserine suggest that both states are present at neutral pH, but the −2 state is predominant[61].

We carried out 250-ns all-atom MD simulations in explicit solvent for each of the selected four models in both unphosphorylated and phosphorylated states, as well as a simulation of the known LC3B-p62 complex as a reference. The MD simulations were performed with Gromacs version 5.1[62] with the protein force fields Charmm22/CMAP[63] and Charmm22*[64] in conjunction with the TIPS3P water model. Parameters for S2P were merged from ref. [65]. All systems were solvated in dodecahedral boxes with a minimum distance between protein and box edge of 1.8 nm and charged ions were added to set a concentration of 0.15 M NaCl. The productive MD simulations were carried out at 298 K with simulation parameters and equilibration protocol[66]. The first 50 ns of each MD simulation required for equilibration of the system was discarded from downstream analyses.

For each trajectory we calculated, as a measure of protein flexibility, the average RMSF for each Cα atom in the Ambra1-LIR or the p62-LIR peptides averaging the RMSF over 10-ns time windows and estimating the associated standard deviations.

The four independent replicates of each system were merged into a concatenated trajectory for the analyses of electrostatic interactions.

We estimated intra- and inter-molecular electrostatic interactions, as implemented in the PyInteraph suite of tools[67]. The interactions are defined as distances between the charged groups in the residue side chains of arginine, lysine, aspartate, glutamate and the phosphoserine. We employed a 5 Å cutoff in the analyses. To filter out interactions with low occurrence in the simulations, we included in the final networks only those interactions which are present in at least

20% of the simulation frames, as previously tested in other cases of study[67]. All the electrostatic interactions identified for each of the MD ensembles were visualized using the PyMOL plugin xPyder[68].

**Mass spectrometry analysis of AMBRA1-interacting proteins**. SILAC-based strategy is used for comparative and quantitative AMBRA1-interacting proteins analysis in HeLa cells in mitophagy condition. Cells were growth in SILAC DMEM (Silantes, 280001200), supplemented with FBS 10%, 1% penicillin–streptomycin solution, L-glutamine, L-arginine and L-lysine, according to the supplier's instructions. Accordingly, we obtained two different isotope amino-acid labelling media: Light Medium K0/R0 and SILAC Heavy medium (K6/R10). Cells were growth in these media for 2 weeks.

In order to analyse AMBRA1 interactors during mitophagy, Light and Heavy cells were transfected with a vector encoding for Myc-AMBRA1^ActA for 24 h. Equal amounts of harvested Light and Heavy lysates were immunoprecipitated with an anti-Myc antibody, as previously described in the immunoprecipitation protocol. Immunoprecipitated heavy proteins were then mixed 1:1 with light proteins and subjected to SDS–PAGE followed by band excision and trypsin-peptide digestion in order to allow LC-ESI-MS/MS (liquid chromatography-electrospray ionization-tandem mass spectrometry) analysis[15].

**In vitro ubiquitylation assay**. HEK293T cells stably expressing Flag-HUWE1 were lysed with Triton Buffer (50 mM Tris-Hcl pH 7.5, 250 mM NaCl, 50 mM NaF, 1 mM EDTA 1 pH 8, 0,1% Triton), supplemented with proteases and phosphatases inhibitors. Then, 3–5 mg of whole cell lysates were precleared with protein A/G-Sepharose beads (GE Healthcare) for 3 h and then incubated overnight at 4 °C with 5 mg of anti-Flag antibody (Sigma-Aldrich, F3165) per sample. Finally, the immunocomplexes have been absorbed on protein A/G-Sepharose beads, washed six times in Triton lysis buffer and used for in vitro ubiquitylation assay or eluted by boiling in SDS loading buffer.

In vitro ubiquitylation assay was carried out using HUWE1 complex immunopurified from HEK293T cells stably expressing Flag-HUWE1 using the method described above. The beads containing the HUWE1 immunocomplexes were washed twice with equilibration buffer (25 mM Tris-Hcl pH 7.5, 50 mM NaCl, 1 mM EDTA, 0.01% Triton and 10% glycerol) and then added to the following reaction: 40 ng of E1, 0.5 µg of UbcH3, 0.5 µg of UbcH5, 0.3 µg of UbcH7, 1 µg of ubiquitin in the ubiquitylation buffer (2.5 mM Tris·HCl, 0.7 mM dithiothreitol (DTT), 4 mM ATP, 10 mM MgCl₂, 0.1 mM ubiquitin aldehyde) in a final volume of 30 µl. The assays have been carried out for 90 min at 30 °C, then the immunocomplexes were eluted by boiling in SDS loading buffer, loaded on SDS–PAGE, blotted onto PVDF membranes (Millipore) and then blocked with PBS-T (phosphate-buffered saline and 0.1% Tween-20) containing 5% non-fat dry milk for 1 h at room temperature (RT). The incubation with primary antibodies was performed for 2 h at RT, followed by appropriate horseradish peroxidase-conjugated secondary antibodies for 1 h at RT. After extensive washing, detection was performed with the ECL Western Blot Chemiluminescence Reagent (Perkin Elmer).

**In vitro protocols of ischaemia**. Preconditioning and ischaemia condition were recreated by exposing cells to 30 min and 3 h, respectively, of OGD[69]. In addition, cells were exposed to 1 h of OGD. Initially, OGD was performed in a medium previously saturated with 95% N₂ and 5% CO₂ for 20 min and containing NaCl 116 mM, KCl 5.4 mM, MgSO₄ 0.8 mM, NaHCO₃ 26.2 mM, NaH₂PO₄ 1 mM, CaCl₂ 1.8 mM, glycine 0.001 mM and 0.001 w/v phenol red[33,70]. Then, hypoxic conditions were maintained using a hypoxia chamber at 37 °C, 5% CO₂ and 95% N₂. At the end of OGD, the cells were incubated for 24 h in normoxic conditions using DMEM containing normal levels of O₂ in a CO₂ incubator at 37 °C RX) followed by lysis for western blot analysis. Whole cell lysate were prepared in RIPA buffer (50 mM Tris-HCl, 150 mM NaCl, 1% (v/v) Triton X-100, pH 8.0) containing protease and the phosphatase inhibitors.

**mt-mKeima mitophagy assay**. Wild-type and PINK1-KO HeLa cells stably expressing mitochondrial-targeted mKeima (mt-mKeima,[7]) were co-transfected with PcDNA3+GFP, Myc-AMBRA1^WT+GFP, Myc-AMBRA1^S1014A+GFP, Myc-AMBRA1^S1014D+GFP, PcDNA3+GFP-ShCtr, Myc-Ambra1^ActA+GFP-ShCtr or myc-Ambra1^ActA+GFP-ShHUWE1, and treated as indicated in the text. GFP was used as a reporter for co-transfected cells. Live cells were photographed at 37 °C/5% CO₂ through the live cell imaging microscope UltraView Vox (Perkin Elmer). mt-mKeima was excited both at 405 nm (neutral, pseudo-coloured in green) and 561 nm (acidic, red) and detected through the same emission filter (615 nm, W70), while GFP was captured with standard procedures (488 nm excitation, 525 nm W50 emission filter, pseudo-coloured in cyan). A total of 100 randomized fields per condition were detected in each experiment with a 100× objective. Randomization was performed through an automated function of the miscroscope. mt-mKeima fluorescence intensity was quantified in single GFP-positive cells with the integrated function of the microscope software Volocity (Perkin Elmer). Fields with no GFP-positive cells were excluded from the analysis.

**Statistical analysis**. All statistical calculations were performed and graphed using GraphPad Prism 6. Comparisons between two groups were analysed using Student's t-test assuming a two-tailed distribution. Three or more group comparisons were performed with one-way analysis of variance (ANOVA) and Tukey's post-test with a confidence interval of 95%. Significance is defined as *P < 0.05; **P < 0.01; ***P < 0.001; ****P < 0.0001. No statistical methods were used to predetermine sample size.

## Data availability

The data that support the findings of this study are available from the corresponding author upon reasonable request.

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

## Acknowledgements

We wish to thank: R. Youle (Bethesda, MD) for kindly providing us Penta-KO and mt-mKeima expressing wild-type or PINK1 knockout HeLa cells; T. Mak (Toronto, CDN) for HUWE1 overexpressing cells; E. Meylan (Lausanne, CH), M. Karin (La Jolla, CA) and I. Dikic (Frankfurt, D) for supplying us the constructs Flag-IKKα^WT, HA-IKKα^K44M and Myc-TBK1, respectively. We thank Elsevier for giving us the opportunity to re-use on Fig. 8 our previous mitochondria draw image, published in Strappazzon and Cecconi. We are grateful to the Danish Supercomputing Center Computerome. This work was supported by grants: Italian Ministry of Health GR2011-02351433 and "Roche per la Ricerca 2017" to F.S.; AIRC (IG2016–18906), KBVU from the Danish Cancer Society (R72-A4408, R146-A9364), the Novo Nordisk Foundation (7559, 22544), the European Union (Horizon 2020 MEL-PLEX, grant agreement 642295), the Lundbeckfonden (R233-2016-3360), the LEO Foundation (LF17024), the Bjarne Saxhof Foundation to F.C.; PRACE-DECI14th GRANT for HPC resources on Archer (UK) to E.P. Further, F.C. and E.P. labs in Copenhagen are part of the Center of Excellence in Autophagy, Recycling and Disease (CARD), funded by the Danish National Research Foundation; and AIRC (MFAG#15523) to A.P. The work of J.G., V.D. and V.V.R. was supported by the German Research Foundation (CRC/SFB 1177 on selective autophagy). Research conducted by D.S. and M.C. was supported by the Umberto Veronesi Foundation, LAM-Bighi Grant Initiative, FIRB-Research Consolidator Grant [RBFR13P392] and Italian Ministry of Health [IFO14/01/R/52]. We thank M. Acuña Villa for secretarial work and S. Verna and N. Rogova for technical assistance.

## Author contributions

A.D.R. performed most experiments and designed the work plan together with F.S. and F.C. Experimental work: F.S. (cloning of serine mutants); A.P. (IVT ubiquitylation); M.

N., M.L., E.P. (molecular modelling of AMBRA1–LC3); J.G., V.D., V.V.R. (ITC and NMR); Z.H. and J.D. (mass spectrometry analysis); S.E.A. (P-S1014-AMBRA1 antibody generation); P.D.A. (mito-mKeima experiments); D.S. and M.C. (in vitro ischaemic experiments); and G.M. (contribution of reagents). A.D.R., F.S. and F.C. wrote the manuscript.

## Additional information

**Competing interests:** The authors declare no competing interests.

