## [Peer Review File · Nature Communications]

Reviewers' comments:

Reviewer #1 (Remarks to the Author):

The manuscript reports the identification and elucidation of two key interactions of AMBRA1 in the regulation of PINK1/PARKIN-independent mitophagy (degradation of mitochondria via autophagy) via extensive cell culture based experiments: i) AMBRA1 recruits the E3 ubiquitin ligase HUWE1 to mitochondria to promote the ubiquitination and degradation of the outer mitochondrial membrane protein MNF2, thereby promoting mitophagy, and ii) AMBRA1 is phosphorylated by the kinase IKK α to promote binding of AMBRA1 to the GABARAP/LC3 family of autophagy-related proteins, thereby targeting the mitochondrion for autophagosomal degradation. The interaction between AMBRA1 and GABARAP/LC3 proteins is further characterized biophysically using ITC and NMR spectroscopy.

I find the evidence presented in the paper clear and convincing. The resulting conclusions are of sufficient novelty and of considerable biological interest, and in the discussion the authors point out (convincingly again) that the study opens up several avenues for follow-up studies geared at understanding the regulation of mitophagy. I would therefore recommend publication of this manuscript in Nature Communications after minor revisions, as presented below.

As pointed out already, I find the scientific evidence convincing overall. Given that the authors discuss at length that both, AMBRA1 and HUWE1, have been reported to interact with c-Myc, I am wondering whether the extensive use of Myc-tagged constructs and corresponding antibodies could cause any complications or artifacts. Please discuss this issue (ideally also in the manuscript itself). On a different note, I am wondering whether the conclusion that phosphorylation at S1014 causes a "conformational change" in AMBRA1 is really the right expression that intuitively describes the results presented in Fig. 4e most accurately. On a first glance it looks more like a reduction in heterogeneity (yes, that technically qualifies as a "conformational change" as well, in a very general sense), or in other words, a stabilization of a subset of preferred conformations, rather than a defined change from one conformation to another. Maybe the authors could present this a little more clearly. Also, which protonation state of the phosphoserine residue was used in the MD simulations and why (pKa/pH)? And why is the discussion of the interaction in the main text focused almost exclusively on LC3B whereas the GABARAP data is banned into SI, when in fact the interactions with GABARAP are significantly stronger?

The presentation of the manuscript still has minor potential for improvements. The style of the results section and the figure legends is extremely schematic but easy to follow so I am fine with that. The English is also easy to read but a little rough at times (e. g., "capable of inducing", p. 7; "Of the highest importance"?, p. 12; "this evidence", p. 14; "neurodegenerative disease", p. 15; the occasional missing article; a/an is based on pronunciation not spelling – "a ubiquitylation cascade" but "an NF κ B"; "either ... or...", SI; etc.). Please proofread it one more time, maybe with the help of a native speaker. Also, I think some of the figure references are not up to date any more: I could be wrong but I suspect that Fig. 1j are the cells of Fig. 1i?

The figure legend to Fig. 4e/f is definitely insufficient. What are C22/C27 (I suspect that refers to a particular version of the CHARMM force field)? What are these red tubes (perhaps some way of presenting salt bridges)? What does the green color signify?

As noted above, I think that some of the GABARAP stuff from the SI should probably be mentioned in the main text, at least briefly, including the low enthalpy of binding. If necessary, it might be possible to move Tables 1 to 3 to SI if space is at a premium.

Reviewer #2 (Remarks to the Author):

In this study, the authors have explored the mechanisms by which AMBRA1 regulates mitophagy in cells. There are several new and potentially intriguing findings in this study. First, the authors' data implicate the E3 ubiquitin ligase HUWE1 as a regulator of AMBRA1-mediated mitophagy. Second, phosphorylation of AMBRA1 by IKK α functions to increase the interaction between AMBRA1 and LC3. The authors also demonstrate that HUWE1 is required for IKK α to

phosphorylate AMBRA1 and induce mitophagy. Although this is an interesting study, there are a number of concerns with the study. Major concerns include lack of appropriate controls in many of the experiments and the low sample number. Detailed concerns are discussed in detail below.

-All the studies on AMBRA1 in this manuscript are based on overexpression of proteins or chemically induced mitophagy using mitochondrial toxin. Although the findings that HUWE1 and IKKalpha regulate AMBRA1-mediated mitophagy are pretty convincing, it is still unclear under what physiological conditions this pathway is activated. It will be important for the authors to confirm that this is a physiologically relevant pathway that they are studying and that it is activated in response to a physiological stress/challenge other than mitochondrial poisons.

- A major concern is the lack of important controls in many of the experiments involving western blotting of transfected cells.

- Experiments that lack appropriate controls:

a. Figure 1h lacks a baseline controls (i.e. Myc vector alone). Without the control, it is unclear how much mito-targeted AMBRA1 enhances ubiquitination of mitochondrial proteins.

b. Figure 2c needs a pcDNA3 + O/A control. Without this control, it will not be known whether the effect on mitochondria by O/A exposure is specifically due to AMBRA1 overexpression.

c. Figure 3b - The co-IP lack control samples without O/A treatment. These need to be included.

d. Figure 3c lacks control transfected cells. It will be very important to include a control here because it looks like just overexpressing AMBRA1 alone activates mitophagy (even with the S1014A mutant). Unless a control is included it is unknown how mitochondrial content is altered.

e. Figure 3d - Imaging experiment needs a control to compare with. Also, the authors state that the AMBRA1S1014A mutant causes mitochondrial clustering but the images shows the opposite.

f. Figure 5d - control is missing. Need to include a non O/A condition with the IKKalpha.

g. Figure 5h - need a Bay alone + AMBRA1 control in this experiment.

- Statistical analysis - The sample number analyzed is also very low varies from n=1 to n=3. Also, the authors have analyzed a lot of their data using one way ANOVA. Although the ANOVA test informs whether you have an overall difference between your groups, but it does not tell you which specific groups differed. For this a post hoc test must be done. However, the authors have not listed which post hoc test was used to compare the groups.

- Figure 1k - what happens to other mitochondrial proteins? Are they also changed like Mfn2?

- Is phosphorylated AMBRA1 selectively associated with depolarized mitochondria?

-Many of the "representative blots" shown are not very representative of what is shown in the bar graph. In many blots, it looks like the mitochondrial proteins are not changing but the bar graph shows sometimes over a 50% change. For example, see blots and bar graphs in Figures 1f, 5g, 5h 5i, and 5j (plus their corresponding supplemental). This is very concerning since these experiments were only repeated 3 times.

-The authors previously reported that AMBRA1 acts as a mitophagy receptors and interacts with LC3 via its LIR (Strappazzon et al Cell Death Diff. 2015) so that aspect of the manuscript is not particularly novel. Instead, it is recommended that the authors focus on the new findings. Thus, sections stating that "we thus found that AMBRA1 is a novel mitophagy receptor" should be revised since that was not discovered in the current study.

-The authors need to tone down their interpretation of the MFN2 data or perform additional experiments to support their claims. For instance, they conclude that HUWE1 is critical in AMBRA1-mediated mitophagy by promoting degradation of MFN2. However, the authors have not demonstrated that Mfn2 degradation is required for AMBRA1-mediated mitophagy, just that it is degraded in a HUWE1-dependent manner. In fact, most of Mfn2 is still degraded even when HUWE1 is knocked down (see Figure 1k). Thus, the authors need to revise the first and last paragraph of the discussion (plus the last paragraph in the introduction).

There are numerous typos throughout the manuscript. A spellcheck needs to be performed.

-Olygomycin is spelled oligomycin

Reviewer #3 (Remarks to the Author):

This manuscript describes that E3 ligase HUWE1 and IKK α kinase regulate AMBRA1-mediated mitophagy, (1) HUWE1 interacts and collaborates with AMBRA1 to induce MFN2 degradation and mitophagy, (2) AMBRA1 may be a novel mitophagy receptor, (3) AMBRA1 phosphorylation on Serine S1014 induces a conformational change and promotes its interaction with LC3/GABARAP, (4) AMBRA1 phosphorylation on Serine S1014 is mediated by IKK α and (5) HUWE1 regulates AMBRA1 phosphorylation on Serine S1014 following mitophagy induction, consequently AMBRA1, HUWE1 and IKK α regulates mitophagy and mitochondrial clearance. This study reveals a novel pathway of mitophagy and mitochondrial homeostasis. However, there are several points need to be addressed more clearly and improved.

Major points:

1. The authors have found that HUWE1 interacts with AMBRA1 and ubiquitylates MFN2, however, they did not mention if AMBRA1 is the substrate of HUWE1 or AMBRA1 is regulated by HUWE1 through ubiquitination.
2. AMBRA1 LIR motif and adjacent region is responsible for LC3 interaction and phosphorylation, which region is required for HUWE1 and AMBRA1 interaction? AMBRA1 phosphorylation on Serine S1014 by IKK α is crucial for AMBRA1-induced mitophagy, however, is it important for HUWE1-AMBRA1 interaction and MFN2 degradation?
3. The authors show that HUWE1 controls AMBRA1-induced mitophagy, does HUWE1 affect mitophagy under the condition without AMBRA1 overexpression or other kinds of mitophagy?
4. The authors have shown that AMBRA1 phosphorylation on Serine S1014 regulates AMBRA1 and LC3 interaction, and HUWE1 regulates AMBRA1 phosphorylation, does HUWE1 affect AMBRA1 and LC3 interaction? Do HUWE1 and IKK α affect each other during AMBRA1-dependent mitophagy?
5. This manuscript claims that HUWE1 promotes PINK1/PARKIN-independent mitophagy by regulating AMBRA1 activation through IKK α , however, all experiments of AMBRA1-induced mitophagy use cells with PINK1/PARKIN expression, some PINK1/PARKIN-deficient cells may be required to further confirm this point.
6. Only indirect assays were used to measure mitophagy in this study, more direct assay would be helpful.

Minor points,

1. Page 4, the second sentence, ref.15 does not mention PINK1/PARKIN-independent mitophagy, it is ref.16.
2. Page 7 and Supplementary Fig. 2b, NH₄Cl not only inhibits lysosomal degradation, but also affect endosomal and cytosolic pH, more specific lysosomal inhibitor should be used.
3. Page 24, in vitro ubiquitination assay, the substrate is not described. In Figure 1j legend, more detail should be added, for example, what kind of antibody was used, anti-Ub or anti-MFN2 antibody?
4. Page 29, Figure 6 legend, no result supports ".....HUWE1 at the mitochondria....." in figure 6.
5. Figure 5d, this figure is confused, why is BAY-117082 added here?
6. In Figure 5h and 5j, Supplementary Figure 5e, only slight rescue is shown and they does not support the conclusion in context. In Supplementary Figure 5a, the quality of P-S1014-AMBRA1 image need to be improved.

Reviewers' comments:

Reviewer #1 (Remarks to the Author):

The manuscript reports the identification and elucidation of two key interactions of AMBRA1 in the regulation of PINK1/PARKIN-independent mitophagy (degradation of mitochondria via autophagy) via extensive cell culture based experiments: i) AMBRA1 recruits the E3 ubiquitin ligase HUWE1 to mitochondria to promote the ubiquitination and degradation of the outer mitochondrial membrane protein MNF2, thereby promoting mitophagy, and ii) AMBRA1 is phosphorylated by the kinase IKK α to promote binding of AMBRA1 to the GABARAP/LC3 family of autophagy-related proteins, thereby targeting the mitochondrion for autophagosomal degradation. The interaction between AMBRA1 and GABARAP/LC3 proteins is further characterized biophysically using ITC and NMR spectroscopy.

I find the evidence presented in the paper clear and convincing. The resulting conclusions are of sufficient novelty and of considerable biological interest, and in the discussion the authors point out (convincingly again) that the study opens up several avenues for follow-up studies geared at understanding the regulation of mitophagy. I would therefore recommend publication of this manuscript in Nature Communications after minor revisions, as presented below. As pointed out already, I find the scientific evidence convincing overall. Given that the authors discuss at length that both, AMBRA1 and HUWE1, have been reported to interact with c-Myc, I am wondering whether the extensive use of Myc-tagged constructs and corresponding antibodies could cause any complications or artifacts. Please discuss this issue (ideally also in the manuscript itself).

We thank the reviewer for his/her very positive consideration. In order to demonstrate that AMBRA1-HUWE1 interaction is not influenced by the Myc-tag of AMBRA1, we transfected HeLa cells with a construct encoding an untagged-AMBRA1 and treated them with the O/A combined treatment (2.5 μ M, 0.8 μ M, 1h). As shown in the Figure A here below, we observed that untagged-AMBRA1 still interacts with HUWE1 in normal conditions and that this binding is enhanced following mitophagy induction.

Figure A: HeLa cells were transfected with a vector encoding AMBRA1 and then treated with O/A for 1h. Lysates were subjected to immunoprecipitation by using an anti-HUWE1 antibody in order to precipitate HUWE1 or IgG as a control. Immunocomplexes were analysed using anti-AMBRA1 and anti HUWE1 antibodies.

On a different note, I am wondering whether the conclusion that phosphorylation at S1014 causes a "conformational change" in AMBRA1 is really the right expression that intuitively describes the results presented in Fig. 4e most accurately. On a first glance it looks more like a reduction in heterogeneity (yes, that technically qualifies as a "conformational change" as well, in a very general sense), or in other words, a stabilization of a subset of preferred conformations, rather

than a defined change from one conformation to another. Maybe the authors could present this a little more clearly.

We agree that the formulation *conformational change* could in principle be misleading for the reader, since it could be interpreted as the change from one conformation to another. We thank the referee for this suggestion and we clarified our formulation in the new version of the manuscript (abstract, results, figure legend: line 38 page 1; line 15 page 11; line 28 page 30).

Also, which protonation state of the phosphoserine residue was used in the MD simulations and why (pKa/pH)?

We thank the reviewer for this nice catch. The protonation state of the phosphoserine (charge -2) has been added to the methods section (line 12 page 24). We modelled the phosphoserine S1014 using the -2 state since the pKa values of single protonated/un-protonated transition of phosphoserine suggest that both states are present at neutral pH, but the -2 state will be predominant accordingly to [Kitchen J, Saunders RE, Warwicker J. Charge environments around phosphorylation sites in proteins. *BMC Struct Biol.* 2008;8:19, Andrew CD, Warwicker J, Jones GR, Doig AJ. Effect of phosphorylation on alpha-helix stability as a function of position. *Biochemistry.* 2002;41:1897–1905]. These papers are now cited in the revised text.

And why is the discussion of the interaction in the main text focused almost exclusively on LC3B whereas the GABARAP data is banned into SI, when in fact the interactions with GABARAP are significantly stronger?

We kindly appreciate this reviewer suggestion and his consideration on GABARAP data. However, most of the work has been performed analysing the LC3B/AMBRA1 binding in order to study in depth the interaction described previously by Strappazzon and coll. During our analysis, we decided to include also GABARAP proteins in order to verify whether AMBRA1 phosphorylation was important for its interaction with other mATG8s. Unexpectedly, we obtained the data indicating that AMBRA1/GABARAP interaction is stronger than LC3B/AMBRA1 binding; further studies (by biochemistry and microscopy analyses) will be necessary to confirm this result, obtained only by a structural approach.

The presentation of the manuscript still has minor potential for improvements.

The style of the results section and the figure legends is extremely schematic but easy to follow so I am fine with that.

The English is also easy to read but a little rough at times (e. g., "capable of inducing", p. 7; "Of the highest importance"?, p. 12; "this evidence", p. 14; "neurodegenerative disease", p. 15; the occasional missing article; a/an is based on pronunciation not spelling – "a ubiquitylation cascade" but "an NFkappaB; "either ... or...", SI; etc.). Please proofread it one more time, maybe with the help of a native speaker.

Thank you very much for the suggestions. We revised the manuscript to ameliorate the language use.

Also, I think some of the figure references are not up to date any more: I could be wrong but I suspect that Fig. 1j are the cells of Fig. 1i?

Nice catch again, we corrected the legend of Figure 1j accordingly.

The figure legend to Fig. 4e/f is definitely insufficient. What are C22/C27 (I suspect that refers to a particular version of the CHARMM force field)? What are these red tubes (perhaps some way of presenting salt bridges)? What does the green color signify?

Legend of figure 4e/f has been revised. Furthermore, numbering on the x-axis of figure 4e has also been corrected.

As noted above, I think that some of the GABARAP stuff from the SI should probably be mentioned in the main text, at least briefly, including the low enthalpy of binding. If necessary, it might be possible to move Tables 1 to 3 to SI if space is at a premium.

We thank the reviewer for his suggestion. In the present reviewed version we supplement the main text with data of GABARAB.

Reviewer #2 (Remarks to the Author):

In this study, the authors have explored the mechanisms by which AMBRA1 regulates mitophagy in cells. There are several new and potentially intriguing findings in this study. First, the authors' data implicate the E3 ubiquitin ligase HUWE1 as a regulator of AMBRA1-mediated mitophagy. Second, phosphorylation of AMBRA1 by IKKalpha functions to increase the interaction between AMBRA1 and LC3. The authors also demonstrate that HUWE1 is required for IKKalpha to phosphorylate AMBRA1 and induce mitophagy. Although this is an interesting study, there are a number of concerns with the study. Major concerns include lack of appropriate controls in many of the experiments and the low sample number. Detailed concerns are discussed in detail below.

-All the studies on AMBRA1 in this manuscript are based on overexpression of proteins or chemically induced mitophagy using mitochondrial toxin. Although the findings that HUWE1 and IKKalpha regulate AMBRA1-mediated mitophagy are pretty convincing, it is still unclear under what physiological conditions this pathway is activated. It will be important for the authors to confirm that this is a physiologically relevant pathway that they are studying and that it is activated in response to a physiological stress/challenge other than mitochondrial poisons.

We thank the Referee for this extremely useful comment that drives us to significantly improve the robustness of our key conclusions. Indeed, we decided to now analyse AMBRA1-mediated mitophagy under a particular form of mitochondrial physiological stress due to ischemia. In fact, in this revision we added new collaborators, who previously demonstrated that mitophagy is induced upon ischemic stimuli (Matic et al., 2016). We thus checked with their help our model in the neuroblastoma cell line SHSY5Y, widely used under ischemic conditions (Matic et al., 2016; Zhou et al., 2016; Xiao et al., 2017). Since we found that AMBRA1 is phosphorylated upon mitophagy induction (both following FCCP and OA treatments), we decided to analyse (novel Figure 7A) the phosphorylation status of AMBRA1 at S1014 in this more physiological model of mitophagy. We thus performed a western blot analysis, in which we analysed AMBRA1-phosphorylation at 30 min, 1 hour and 3 hours of OGD/RX. As observed in figure 7A, AMBRA1 is phosphorylated between 30' and 1 hour of ischemic stimuli, confirming its mitophagic activation.

Next, we analysed the biological relevance of *endogenous* AMBRA1 in ischemic-mitophagy. To this end, we downregulated *endogenous* AMBRA1 in SHSY5Y cells exposed to oxygen/glucose deprivation for 3 hours followed by re-oxygenation for 24 hours (OGD/RX). As expected, we observed a reduction of COXII mitochondrial marker in the control conditions (scrambled siRNA), upon ischemic stroke; by contrast, the mitochondrial protein level was completely restored when we downregulated AMBRA1 (Figure 7B). This result suggests that endogenous AMBRA1 has a prominent role in ischemic-induced mitophagy in SHSY5Y cells. Finally, we decided to analyse the

endogenous role of HUWE1 and IKK α in AMBRA1-mediated mitophagy also in this system. In Figure 7C, AMBRA1^{WT} expressing SHSY5Y cells were downregulated for HUWE1 by siRNA, and next exposed to 3hrs of OGD/RX. As shown in the western blot, AMBRA1 expression is sufficient to reduce the levels of COXII that were rescued by SiRNA-HUWE1 transfection, as previously obtained in HeLa cells, following O/A treatments (see Figure 1g). To strengthen these data, we analysed the role of IKK α by expressing the dominant negative form IKK α ^{K44M} in AMBRA1^{WT}-expressing SHSY5Y cells subjected to 3hrs of OGD/RX. Again, we obtained a block in mitochondrial clearance by using the dominant form of IKK α . All together these new findings suggest that AMBRA1 can regulate ischemic mitophagy in a novel pathway, together with HUWE1 and IKK α .

A major concern is the lack of important controls in many of the experiments involving western blotting of transfected cells.

-Experiments that lack appropriate controls:

a. Figure 1h lacks a baseline controls (i.e. Myc vector alone). Without the control, it is unclear how much mito-targeted AMBRA1 enhances ubiquitination of mitochondrial proteins.

We thank the reviewer for his accurate analysis. Admittedly many controls of the previous version of this work could be implemented. We have changed Figure 1h and its relative quantification, including the control condition PcDNA+SiRNA-Ctr.

b. Figure 2c needs a pcDNA3 + O/A control. Without this control, it will not be known whether the effect on mitochondria by O/A exposure is specifically due to AMBRA1 overexpression.

In Figure 2c we used Penta-KO cells described as mitophagy-unable cells in Lazarou et al., 2015. These authors demonstrated that only the expression of the autophagy receptors NDP52 and OPTN (driven by PINK1-overexpression) regulates mitophagy in these cells. However, in order to address the Referee request, we added in the new Supplementary Figure 2a a Western blot analysis of Penta-KO cells treated or not with O/A, which confirms the previous findings of Youle's group.

c. Figure 3b - The co-IP lack control samples without O/A treatment. These need to be included.

In order to address the appropriate Reviewer request, we replaced Figure 3B with a new panel, in which we analysed the interaction between AMBRA1 and LC3B in the presence or absence of the combined O/A treatment. Indeed, we transfected cells with Myc-AMBRA1^{WT}, or Myc-AMBRA1^{S1014A} or Myc-AMBRA1^{S1014D} and treated them or not with O/A. Then, we immunoprecipitated AMBRA1 in the mitochondrial fraction, by using an anti-Myc antibody and we evaluated the immunocomplex formation by western blot. As illustrated in the new Figure 3B, we not only confirm that AMBRA1 interacts with LC3B upon mitophagy stimulation, but also that the AMBRA1-LC3B binding is strongly related to the AMBRA1-S1014 phosphorylation state. In fact, the phospho-dead mutant shows a reduction in LC3B binding, respect to the phospho-mimetic form of AMBRA1.

d. Figure 3c lacks control transfected cells. It will be very important to include a control here because it looks like just overexpressing AMBRA1 alone activates mitophagy (even with the S1014A mutant. Unless a control is included is it unknown how mitochondrial content is altered.

We fully agree and we included the control transfection with the empty vector PcDNA3 in the presence or absence of treatment in the new Figure 3C.

e. Figure 3d – Imaging experiment needs a control to compare with. Also, the authors state that the AMBRA1S1014A mutant causes mitochondrial clustering but the images shows the opposite.

We absolutely agree with the reviewer request and, in order to address this issue, we included the control transfection with the GFP transfected cells in the new Figure 3D; moreover, we added new images for wild-type and S1014-mutants transfected cells. As observed, we obtained a fragmentation of the mitochondrial network in all transfected cells, treated with the mitophagy stimuli O/A. However, we also observed a significant decrease of the area occupied by mitochondria in Myc-AMBRA1^{WT}- and Myc-AMBRA1^{S1014D}-transfected cells, compared to the control (GFP). Finally, as for the sentence “Indeed, we observed that AMBRA1^{S1014A} still induces mitochondria aggregation....” this was a simple mistake for which we apologize. We intended “mitochondrial fragmentation” but in the last version of the manuscript we just removed it.

f. Figure 5d – control is missing. Need to include a non O/A condition with the IKKalpha.

We thank the reviewer for his important suggestion. We replaced Figure 5D with a new western blot analysis and its relative quantification, in which we added also the IKK α transfected sample in the absence of O/A treatment. As shown in the new Figure 5D, AMBRA1 phosphorylation on S1014 is increased when IKK α is expressed, following mitophagy stimulation.

g. Figure 5h – need a Bay alone + AMBRA1 control in this experiment.

Also in this case, we replaced Figure 5H with a novel analysis, in which we added the control Bay alone requested by the Reviewer.

- Statistical analysis –

The sample number analyzed is also very low varies from n=1 to n=3. Also, the authors have analyzed a lot of their data using one way ANOVA. Although the ANOVA test informs whether you have an overall difference between your groups, but it does not tell you which specific groups differed. For this a post hoc test must be done. However, the authors have not listed which post hoc test was used to compare the groups.

We thank the reviewer for this comment, that gives us the possibility to clarify this point and we apologise for the missing information. To assess statistical significance, data from three independent experiments were analysed by using one-way ANOVA and Tukey’s post-hoc test with a confidence interval of 95%. We completed also the MS with this information (See the “Statistical analysis” paragraph).

Further, we improved the number (n) of the majority of the experiments.

- Figure 1k – what happens to other mitochondrial proteins? Are they also changed like Mfn2?

In order to address the referee request, we analysed also the level of the mitochondrial marker COXII at 18 hours post transfection of Myc-AMBRA1^{ActA}. As shown in the Figure B below, the protein levels of COXII do not change at this early step of AMBRA1-induced mitophagy. By contrast, again, Mfn2 levels are downregulated.

Figure B: HeLa cells were transfected with a vector encoding Myc-AMBRA1^{ActA} and downregulated for HUWE1 by SiRNA, as indicated in Figure 1k. Total lysates were then subjected to western blot analysis in order to check the protein level of COXII mitochondrial marker.

- Is phosphorylated AMBRA1 selectively associated with depolarized mitochondria?

We found that AMBRA1 is phosphorylated at S1014 following O/A and FCCP treatments, which are known to induce mitochondria depolarization. By contrast, this phosphorylation event is not observed following autophagy induction (EBSS treatment, figure 3d), this suggesting that this phosphorylation event is, at least, specific following a mitochondrial membrane depolarization event.

In addition, thanks to the reviewer suggestion, we reproduced our results in a more physiological condition, which is characterized by hypoxia-reoxygenation-induced mitochondrial depolarization (Zhang et al., 2013; Khader et al., 2016). In this context, we confirmed that endogenous AMBRA1 is phosphorylated on S1014. We can thus conclude that AMBRA1 phosphorylation on S1014 is associated with depolarized mitochondria following mitochondrial poison treatments or in a more physiological context, such as hypoxia-reoxygenation *in vitro*. This said, we cannot exclude that this post-translational modification could occur in other contexts.

-Many of the “representative blots” shown are not very representative of what is shown in the bar graph. In many blots, it looks like the mitochondrial proteins are not changing but the bar graph shows sometimes over a 50% change. For example, see blots and bar graphs in Figures 1f, 5g, 5h 5i, and 5j (plus their corresponding supplemental). This is very concerning since these experiments were only repeated 3 times.

In order to address the important referee request, we replaced blots with a more representative image for figures 1f, 5i and 5j. Besides, we supplemented an additional experiment for figure 5h. Concerning Figure 5g, we already put the best illustration that reflects the trend of the graph (indeed, one should take also the loading control levels into consideration). Moreover, in general, we pointed out that all the experiments were repeated three or in most cases more times, but we decided to quantify only three experiments in order to uniform the analysis along the paper and because the data are already significant with n=3.

-The authors previously reported that AMBRA1 acts as a mitophagy receptors and interacts with LC3 via its LIR (Strappazzon et al Cell Death Diff. 2015) so that aspect of the manuscript is not particularly novel. Instead, it is recommended that the authors focus on the new findings. Thus, sections stating that “we thus found that AMBRA1 is a novel mitophagy receptor” should be revised since that was not discovered in the current study.

We thank the reviewer for this suggestion and modified the text in order to focus on the new findings.

-The authors need to tone down their interpretation of the MFN2 data or perform additional experiments to support their claims. For instance, they conclude that HUWE1 is critical in AMBRA1-mediated mitophagy by promoting degradation of MFN2. However, the authors have not demonstrated that Mfn2 degradation is required for AMBRA1-mediated mitophagy, just that it is degraded in a HUWE1-dependent manner. In fact, most of Mfn2 is still degraded even when HUWE1 is knocked down (see Figure 1k). Thus, the authors need to revise the first and last paragraph of the discussion (plus the last paragraph in the introduction).

We thank the Reviewer for his/her consideration and revised the manuscript, as indicated.

There are numerous typos throughout the manuscript. A spellcheck needs to be performed.
-Olygomicin is spelled oligomycin

Nice catch! We corrected these typos in the text.

Reviewer #3 (Remarks to the Author):

This manuscript describes that E3 ligase HUWE1 and IKK α kinase regulate AMBRA1-mediated mitophagy, (1) HUWE1 interacts and collaborates with AMBRA1 to induce MFN2 degradation and mitophagy, (2) AMBRA1 may be a novel mitophagy receptor, (3) AMBRA1 phosphorylation on Serine S1014 induces a conformational change and promotes its interaction with LC3/GABARAP, (4) AMBRA1 phosphorylation on Serine S1014 is mediated by IKK α and (5) HUWE1 regulates AMBRA1 phosphorylation on Serine S1014 following mitophagy induction, consequently AMBRA1, HUWE1 and IKK α regulates mitophagy and mitochondrial clearance. This study reveals a novel pathway of mitophagy and mitochondrial homeostasis. However, there are several points need to be addressed more clearly and improved.

Major points:

1. The authors have found that HUWE1 interacts with AMBRA1 and ubiquitylates MFN2, however, they did not mention if AMBRA1 is the substrate of HUWE1 or AMBRA1 is regulated by HUWE1 through ubiquitination.

We thank the reviewer for this interesting question. In our work, we presented AMBRA1 as a co-factor for HUWE1 E3 ubiquitin ligase activity. We confirmed the MS data by AMBRA1-HUWE1 co-immunoprecipitation and we observed that AMBRA1 is important for HUWE1 translocation to the mitochondria and also for the degradation of its mitochondrial well known substrate MFN2. Since the interaction between E3 ligases and their substrate(s) is weak or transient, it is necessary to add the proteasome inhibitor MG132 in the immunoprecipitation in order to clearly detect the binding. In fact, as for the MFN2 experiment shown in Figure 1i, we added MG132 in order to appreciate the binding between HUWE1 and its substrate. The AMBRA1-HUWE1 interaction, instead, is well visible independently of any proteasome inhibitors (Figure 1B and 1C), thus suggesting that this interaction is not a weak interaction and it is independent from proteasome inhibitor. Since, comparing the binding between AMBRA1 and HUWE1 in the presence or absence of MG132, we did not observe any differences, we can speculate that HUWE1 interacts with AMBRA1 independently of its E3 Ubiquitin ligase activity. Moreover, in order to ascertain that

HUWE1 could not control AMBRA1 protein stability, we performed additional experiments in which we down-regulated HUWE1 by siRNA and analysed AMBRA1 protein levels in the presence or absence of the mitochondria uncouplers O/A (Figures C and D next page, respectively). As shown below, HUWE1 does not regulate AMBRA1 protein stability both in basal or in mitophagy conditions. We thus can confirm that AMBRA1 does not seem to be a substrate of HUWE1. Altogether these results show that the putative AMBRA1-HUWE1 interplay is not based on HUWE1 E3 ligase activity; indeed, this would have been the most obvious regulatory link between these two key factors in mitophagy regulation. This not being the case, and since any further regulation would need further extensive investigation (see also next point), we believe that this task would go beyond the scope of this paper and hope that the Referee will share our view.

Figure C: HeLa cells were transfected with a SiRNA against HUWE1. Total lysates were subjected to western blot analysis in order to check the protein levels of AMBRA1.

Figure D: HeLa cells were transfected with a siRNA against HUWE1 and treated with O/A. Total lysates were subjected to western blot analysis in order to check the protein levels of AMBRA1, upon mitophagy stimulation.

2. AMBRA1 LIR motif and adjacent region is responsible for LC3 interaction and phosphorylation, which region is required for HUWE1 and AMBRA1 interaction?

We thank the Reviewer for this interesting question. In order to analyse the AMBRA1-HUWE1 binding site, we took advantage of three Myc-tagged plasmids encoding several fragments of AMBRA1: F1, F2 and F3, as reported in Fimia et al., 2007.

Indeed, we transfected cells with these plasmids and we induced mitophagy with O/A. Then total lysates were immunoprecipitated by using an anti-HUWE1 antibody and the co-immunoprecipitation was analysed through western blot. As shown in Figure E, HUWE1 binds preferentially the F3 fragment of AMBRA1 and, at a lower extent, its F1 fragment.

Figure E: HeLa cells were transfected with vectors coding for three different Myc-tagged fragments of AMBRA1 (F1, F2, F3). Following O/A treatment (2.5 μ M, 0.8 μ M O/A, 1 hr), endogenous HUWE1 was immunoprecipitated and the complex formation was evaluated by western blot analysis.

Since AMBRA1 and HUWE1 are BH3-containing proteins (Strappazon et al., 2016; Zhong et al., 2005) and since the BH3 domain of AMBRA1 is localized in its C-terminal portion (F3 fragment) we hypothesized that the interaction between AMBRA1 and HUWE1 could be mediated by the BH3 domain of AMBRA1. For this reason, we generated mutant forms of AMBRA1^{ActA} and AMBRA1^{WT} lacking their BH3 domain (AMBRA1^{ActA- Δ BH3} and AMBRA1^{WT- Δ BH3}). In Figure F, we analysed by western blot analysis the immunocomplex formation between these AMBRA1 mutants and HUWE1. As shown in Figure F, AMBRA1 mutants lacking their BH3 domain are still able to interact with HUWE1 (AMBRA1^{ActA- Δ BH3} left panel; AMBRA1^{WT- Δ BH3} right panel). Interestingly, AMBRA1 sequence is characterised by WD domains localized in F1 and F3 fragments. Since WD regions function as a rigid scaffold for protein-protein interactions (Stirnemann et al., 2010), we do not exclude that the AMBRA1-HUWE1 interaction could be due to these domains.

Moreover, in the future, it would be useful to delete the BH3 domain of HUWE1 in order to verify whether this region is necessary for the binding with AMBRA1.

Figure F: HeLa cells were transfected with vectors encoding Myc-AMBRA1^{ActAΔBH3} or Myc-AMBRA1^{WTΔBH3}, left and right panel respectively). Upon mitophagy stimulation, *endogenous* HUWE1 was immunoprecipitated and the complex formation was evaluated by western blot analysis.

AMBRA1 phosphorylation on Serine S1014 by IKK α is crucial for AMBRA1-induced mitophagy, however, is it important for HUWE1-AMBRA1 interaction and MFN2 degradation?

We evaluated AMBRA1-HUWE1 interaction in the presence of the IKK α dominant negative (IKK α K44M). As shown in Figure G, AMBRA1 is still able to interact with HUWE1 even if the phosphorylation event mediated by IKK α is abrogated.

Moreover, as requested, we analysed the MFN2 protein level in AMBRA1 transfected cells (18 hours) in the presence of IKK α ^{K44M}. As shown in the previous version of the manuscript, the HUWE1 downregulation induced a recovery of MFN2 protein levels, but we also observed a recovery of MFN2 levels in IKK α ^{K44M} transfected cells (Figure H, low panel). It thus seems that IKK α could be involved in the turnover of MFN2, maybe mediating a phosphorylation necessary for its degradation and in analogy with the JNK kinase (Leboucher et al., 2012). In addition, since the turnover of MFN2 is highly oscillatory, we cannot exclude that IKK α could also control lysosomal degradation of MFN2. Further studies are necessary to better clarify this function.

Figure G: HeLa cells expressing Myc-AMBRA1^{WT} and treated with O/A. Total lysates were immunoprecipitated with anti-HUWE1 antibody. The immune-complex formations were evaluated by western blot analysis.

Figure H: HeLa cells were transfected with a vector encoding Myc-AMBRA1^{ActA} in combination or not with the kinase dead IKKα^{KM} for 18 hours. Total lysates were subjected to western blot analysis in order to check the protein level of MFN2.

3. The authors show that HUWE1 controls AMBRA1-induced mitophagy, does HUWE1 affect mitophagy under the condition without AMBRA1 overexpression or other kinds of mitophagy?

In order to address the important Reviewer question, we analyse HUWE1 ability to regulate mitophagy in the absence of AMBRA1 over-expression and since HEK293 is a cell line that expresses relatively high levels of Parkin (Pawlyk et al., 2003), we chose this system to analyse a putative role of HUWE1 in another mitophagy pathway (Parkin-dependent). HEK293T and HEK293T stably expressing HUWE1 cells were thus treated with O/A in order to induce mitophagy and total lysates were subjected to western blot analysis. As shown in Figure I, as expected, we observed a significant decrease of mitochondrial marker COXII in HEK293T cells upon mitophagy stimulation. Moreover, this decrease of COXII upon O/A was higher in cells expressing HUWE1, indicating that HUWE1 could affect mitophagy in the absence of AMBRA1 overexpression. Since it is known that AMBRA1 is involved in Parkin-mediated mitophagy (Van Humbeeck et al 2011), we do not know whether HUWE1 could be involved in another kind of mitophagy than endogenous-AMBRA1 dependent mitophagy; however, it can at least stimulate Parkin-mediated mitophagy. Further studies are required to understand whether HUWE1 could be involved in other mitophagy systems (ie: NIX or FUNDC1-dependent mitophagy).

Figure I: HEK293T and HEK-293T-Flag-HUWE1 cells were treated with O/A in order to induce mitophagy. COXII protein levels were analysed by western blot analysis.

4. The authors have shown that AMBRA1 phosphorylation on Serine S1014 regulates AMBRA1 and LC3 interaction, and HUWE1 regulates AMBRA1 phosphorylation, does HUWE1 affect AMBRA1 and LC3 interaction?

In order to solve this question, we performed additional experiments, in which we down-regulated HUWE1 by SiRNA, and we analysed *endogenous* AMBRA1-LC3B interaction by co-immunoprecipitation upon mitophagy stimuli. As shown in Figure 6B, we observed that, upon mitophagy stimulation, AMBRA1-LC3B binding is reduced by silencing HUWE1.

Do HUWE1 and IKK α affect each other during AMBRA1-dependent mitophagy?

In order to address the important referee request, we analysed IKK α protein levels in Si-RNA-Ctr or siRNA-HUWE1 transfected cells. As shown in Figure J, HUWE1 depletion has no effect on IKK α stability upon AMBRA1-induced mitophagy.

Figure J: HeLa cells were transfected with vector encoding Myc-AMBRA1^{ActA} in order to induce mitophagy. Then HUWE1 silencing was obtained by SiRNA transfection and total lysates were subjected to western blot analysis in order to analyse IKK α protein levels.

In order to investigate whether HUWE1 could be affected by the kinase IKK α , we reasoned that IKK is a kinase and thus checked for putative phosphorylation sites in the HUWE1 sequence by using the GPS-software. We found several phosphorylation predictable sites on HUWE1 that could be phosphorylated by IKK α . Although the question is extremely interesting, additional extensive experiments are necessary to analyse whether IKK α could affect HUWE1 by phosphorylation thus controlling AMBRA1-mediated mitophagy. We hope that the Reviewer will share with us the view that these studies will go beyond the scope of this present manuscript.

5. This manuscript claims that HUWE1 promotes PINK1/PARKIN-independent mitophagy by regulating AMBRA1 activation through IKK α , however, all experiments of AMBRA1-induced mitophagy use cells with PINK1/PARKIN expression, some PINK1/PARKIN-deficient cells may be required to further confirm this point.

We thank the reviewer for having given us the possibility to clarify this point. HeLa cells lack *endogenous* PARKIN expression (Denison et al., 2003). To be sure about this, we had checked, before our mass spectrometry analysis, the PARKIN protein levels in HeLa cells, as shown in Figure K here below.

Figure K: Total lysates of HeLa and Hek293 cells were subjected to western blot analysis in order to evaluate endogenous PARKIN level in these two cell lines. We added a control sample in which Hek293 cells were transiently transfected with a vector coding for PARKIN.

Notably, PARKIN is absent in our mass spectrometry results, indicating that, most likely, endogenous PARKIN is missing in this cellular system.

In addition, we previously demonstrated that AMBRA1 can rescue mitophagy in PINK1-/- MEFs and also in primary human cells from a Parkinson's disease patient with PINK1 loss of function (Strappazon et al., 2015). Altogether, these data demonstrate that AMBRA1 can induce mitophagy without PINK1 and PARKIN.

We decided to skip this from our manuscript because it is just a confirmation of the work of Strappazon and coll., but we reproduced our experiments also in HeLa CRISP/Cas9-modified PINK1-KO cells (Lazarou et al., 2015). As shown in the Figure L below, we transfected PINK1-KO cells with a vector encoding Myc-AMBRA1^{ActA} and then analysed mitochondrial clearance by western blot and confocal microscopy analysis, thus confirming that AMBRA1 expression at the mitochondria is sufficient *per se* to induce mitophagy also in PINK1-KO cells.

Figure L: PINK1-KO cells were transfected with a vector encoding Myc-AMBRA1ActA in order to induce mitophagy. We analysed mitophagy induction by western blot and confocal microscopy.

Moreover, we extended our analysis also to the wild-type form of AMBRA1, as shown in Figure M, below.

Figure M: PINK1-KO cells were transfected with vector encoding Myc-AMBRA1^{WT} and treated with O/A in order to induce mitophagy. We analysed mitophagy induction by western blot and confocal microscopy.

Finally, to reinforce our data, we took advantage of mt-mKeima-PINK1-KO cells (a kind gift from Richard Youle, USA). Indeed, we analysed AMBRA1 mediated mitophagy in mt-mKeima-PINK1-KO cells, in which we transfected AMBRA1^{ActA} and down-regulated HUWE1 by ShRNA. As illustrated in Supplementary Fig. 2c, mt-mKeima engulfment into lysosomes results in a spectral shift owing to low pH in GFP-ShCtr+AMBRA1^{ActA} transfected cells. By contrast, only a low decrease of pH was observed in AMBRA1^{ActA} transfected cells in which HUWE1 is depleted. These data not only clearly confirm our previous observations about the role of HUWE1 in AMBRA1-mediated mitophagy, but also highlight the ability of this pathway to activate mitophagy independently of PINK1. We hope by this mean to have assessed the referee request.

6. Only indirect assays were used to measure mitophagy in this study, more direct assay would be helpful.

In order to address the reviewer request, we analysed the AMBRA1 ability to regulate mitophagy in dependence of its phosphorylation state in mt-mKeima-HeLa cells, kindly provided by Richard Youle. Indeed, mt-mKeima-HeLa cells were co-transfected with GFP (in order to select transfected cells) and Myc-AMBRA1^{WT}, or Myc-AMBRA1^{S1014A}, or Myc-AMBRA1^{S1014D}. Cells were then treated with O/A and analysed by a live imaging microscope. As shown in Supplementary Figure 3a, we

confirmed that the phosphorylation of AMBRA1 at S1014 is essential for AMBRA1 mitophagy activity. In fact, Myc-AMBRA1^{S1014A}-transfected cells show a low spectral shift and therefore less mitochondria in autolysosomes compared with the wild-type or with the phospho-mimetic form of AMBRA1. In addition, as already described in the previous answer, we confirmed by the same technique (but in gene-editing PINK1-KO cells), that HUWE1 is crucial for AMBRA1-mediated mitophagy (Supplementary Figure 1c).

Minor points,

1. Page 4, the second sentence, ref.15 does not mention PINK1/PARKIN-independent mitophagy, it is ref.16.

We corrected the reference accordingly.

2. Page 7 and Supplementary Fig. 2b, NH₄Cl not only inhibits lysosomal degradation, but also affect endosomal and cytosolic pH, more specific lysosomal inhibitor should be used.

We thank the reviewer for his suggestions. However, since we transfected AMBRA1^{ActA} in Penta-KO cells for 24 hours, we need to use a lysosomal inhibitor that could contrast the AMBRA1^{ActA} activity during this time (24h). For this reason we chosen NH₄Cl since Chloroquine and Bafilomycin could have undesired effect especially when they are used for a long time (Boya et al., 2003; Boya et al., 2005; Redmann et al., 2017). Moreover, Bafilomycin has several off-target effects on mitochondria, making it hard to discriminate, which are direct consequences of inhibiting autophagy (Redmann et al., 2017).

3. Page 24, in vitro ubiquitination assay, the substrate is not described.

We included the substrate in the new version of the manuscript.

In Figure 1j legend, more detail should be added, for example, what kind of antibody was used, anti-Ub or anti-MFN2 antibody?

We supplemented Figure1j legend with additional information.

4. Page 29, Figure 6 legend, no result supports “.....HUWE1 at the mitochondria.....” in figure 6.

We apologise for this mistake. We change the sentence in “The E3 ubiquitin ligase HUWE1 is crucial for AMBRA1-S1014 phosphorylation”.

5. Figure 5d, this figure is confused, why is BAY-117082 added here?

We thank the reviewer for his consideration. We decided to eliminate BAY-117082 from this figure 5d, since it is just a confirmation of figure 5c.

6. In Figure 5h and 5j, Supplementary Figure 5e, only slight rescue is shown and they does not support the conclusion in context.

We replaced Figure 5h, 5j and Supplementary Figure 5e with new images and relative quantifications that support our conclusions.

In Supplementary Figure 5a, the quality of P-S1014-AMBRA1 image need to be improved.

We improved the quality of P-S1014-AMBRA1 in Supplementary Figure 5a.

REVIEWERS' COMMENTS:

Reviewer #1 (Remarks to the Author):

The manuscript reports the identification and elucidation of two key interactions of AMBRA1 in the regulation of PINK1/PARKIN-independent mitophagy (degradation of mitochondria via autophagy) via extensive cell culture based experiments: i) AMBRA1 recruits the E3 ubiquitin ligase HUWE1 to mitochondria to promote the ubiquitination and degradation of the outer mitochondrial membrane protein MNF2, thereby promoting mitophagy, and ii) AMBRA1 is phosphorylated by the kinase IKK α to promote binding of AMBRA1 to the GABARAP/LC3 family of autophagy-related proteins, thereby targeting the mitochondrion for autophagosomal degradation. The interaction between AMBRA1 and GABARAP/LC3 proteins is further characterized biophysically using ITC and NMR spectroscopy.

I find the evidence presented in the paper clear and convincing. The resulting conclusions are of sufficient novelty and of considerable biological interest, and in the discussion the authors point out (convincingly again) that the study opens up several avenues for follow-up studies geared at understanding the regulation of mitophagy. In the revised version, the authors have addressed the scientific points raised in the original review to my satisfaction and they have added additional, relevant data (new Fig. 7) investigating the role of AMBRA1 in ischemic mitophagy. From a scientific point of view, I would therefore recommend publication of this manuscript in Nature Communications.

Unfortunately, even at the revised stage the presentation of the manuscript still leaves a lot to be desired, including some rather serious issues (missing/duplicate figures) that should definitely have been caught before sending out this manuscript for review. I really do not think it is my job as a reviewer (volunteering my time) to sort all of this out – I will leave it to the editorial office to decide whether this can be fixed at proof stage or not, and there are 17 (!) authors who can do their due diligence and proofread the manuscript carefully.

Just to mention a few specific points that I have noticed:

- * Why is "serine" capitalized throughout the ms?
- * Is the LIR (LC3-interacting region) motif introduced at all when it suddenly appears on p. 2 (l. 33)?
- * IMPORTANT: Fig. 3e has mysteriously disappeared from the ms.
- * What is "Ctrl" (p. 9)?
- * Fig. 1a: "Ratio H/L": heavy/light?
- * IMPORTANT: Supporting information is duplicated (?) for some reason.
- * Supplementary information: "the P1 induces slower exchange than P0" Am I correct in interpreting that to mean that P1 is no longer in the fast exchange regime but in the fast-to-intermediate regime?
- * The typographical and grammatical mistakes (missing articles, singular/plural, wrong use of "either", etc.) are still too numerous for me to list. As I said, more proofreading is urgently needed, ideally also by a native speaker.

Reviewer #2 (Remarks to the Author):

no additional comments

Reviewer #3 (Remarks to the Author):

All comments and questions are responded appropriately and the manuscript is greatly improved.

REVIEWERS' COMMENTS:

Reviewer #1 (remarks to the Author) :

The manuscript reports the identification and elucidation of two key interactions of AMBRA1 in the regulation of PINK1/PARKIN-independent mitophagy (degradation of mitochondria via autophagy) via extensive cell culture based experiments: i) AMBRA1 recruits the E3 ubiquitin ligase HUWE1 to mitochondria to promote the ubiquitination and degradation of the outer mitochondrial membrane protein MNF2, thereby promoting mitophagy, and ii) AMBRA1 is phosphorylated by the kinase IKKa to promote binding of AMBRA1 to the GABARAP/LC3 family of autophagy-related proteins, thereby targeting the mitochondrion for autophagosomal degradation. The interaction between AMBRA1 and GABARAP/LC3 proteins is further characterized biophysically using ITC and NMR spectroscopy. I find the evidence presented in the paper clear and convincing. The resulting conclusions are of sufficient novelty and of considerable biological interest, and in the discussion the authors point out (convincingly again) that the study opens up several avenues for follow-up studies geared at understanding the regulation of mitophagy. In the revised version, the authors have addressed the scientific points raised in the original review to my satisfaction and they have added additional, relevant data (new Fig. 7) investigating the role of AMBRA1 in ischemic mitophagy. From a scientific point of view, I would therefore recommend publication of this manuscript in Nature Communications. Unfortunately, even at the revised stage the presentation of the manuscript still leaves a lot to be desired, including some rather serious issues (missing/duplicate figures) that should definitely have been caught before sending out this manuscript for review. I really do not think it is my job as a reviewer (volunteering my time) to sort all of this out – I will leave it to the editorial office to decide whether this can be fixed at proof stage or not, and there are 17 (!) authors who can do their due diligence and proofread the manuscript carefully. Just to mention a few specific points that I have noticed:

We thank this Referee for his/her nice words on the clarity and importance of our manuscript and we apologize for some inaccuracies that appeared in the last version of the manuscript.

* Why is "serine" capitalized throughout the ms?

In agreement with the Referee's requests, we replaced all the "Serine" with "serine" in the new version of the manuscript.

* Is the LIR (LC3-interacting region) motif introduced at all when it suddenly appears on p. 2 (l. 33)?

We thank the reviewer for its consideration. We have introduced the spelling "LC3-interacting region" (page 2, line 34).

* IMPORTANT: Fig. 3e has mysteriously disappeared from the ms.

We apologize for our mistake. A correct Figure 3, including the 3e panel is now provided in the new version of the manuscript.

* What is "Ctrl" (p. 9)?

We replaced "Ctrl" with "Ctrl-SiRNA" in page 9, line 23.

* Fig. 1a: "Ratio H/L": heavy/light?

The ratio H/L indicates the ratio of heavy labelled peptides to the remaining non-labelled ones (Ratio H/L) (page 32, line 5).

* IMPORTANT: Supporting information is duplicated (?) for some reason.

We apologise for our mistake. We replaced it with a new Supplementary Information file.

* Supplementary information: "the P1 induces slower exchange than P0" Am I correct in interpreting that to mean that P1 is no longer in the fast exchange regime but in the fast-to-intermediate regime?

We can state that titrations of both non-phosphorylated (P0) and phosphorylated (P1) AMBRA1 peptides induce perturbations of the most LC3/GABARAP resonances according to fast exchange regime (almost no changes in intensity of the resonances upon position change). However, these changes in intensity are more pronounced in the case of P1 titration; thus, we define this small difference as "slower exchange". Some (most relevant) resonances are in different exchange regimes (fast for P0 titrations, fast-to-intermediate for P1 titrations). Therefore, to be in agreement with the definitions used in the field, we changed the text in the Supplementary Note 1 into: "[...] however, the P1 induces perturbations of the key LC3B/GABARAP resonances with fast-to-intermediate exchange regime in contrast to P0 (fast exchange regime, Fig. 4b-4c and Supplementary Fig. 4d-4e)."

* The typographical and grammatical mistakes (missing articles, singular/plural, wrong use of "either", etc.) are still too numerous for me to list. As I said, more proofreading is urgently needed, ideally also by a native speaker.

We apologised for our mistakes. We proofread again the text and we believe is now correct on all aspects.

Reviewer #2 (Remarks to the Author):

no additional comments

Reviewer #3 (Remarks to the Author):

All comments and questions are responded appropriately and the manuscript is greatly improved.

We thank these Referees for the nice acceptance of our work!